**communications** engineering

# Super-Turing synaptic resistor circuits for intelligent morphing wing
Atharva Deo[1], Jungmin Lee[1], Dawei Gao[1], Rahul Shenoy[1], Kevin PT. Haughn [2], Zixuan Rong[1], Yong Hei[1], D. Qiao [1], Tanay Topac [3], Fu-Kuo Chang[3], Daniel J. Inman [2] & Yong Chen [1] ✉

Neurobiological circuits in the brain, operating in Super-Turing mode, process information while simultaneously modifying their synaptic connections through learning, allowing them to dynamically adapt to changes. In contrast, artificial intelligence systems based on computers operate in Turing mode and lack the ability to concurrently infer and learn, making them vulnerable to failure under dynamically changing conditions. Here we show a synaptic resistor circuit that operates in Super-Turing mode, enabling concurrent learning and inference. The circuit controls a morphing wing to reduce its drag-to-lift force ratio and recover from stalls in complex aerodynamic environments. The synaptic resistor circuit demonstrates superior performance, faster learning speeds, enhanced adaptability, and reduced power consumption compared to artificial neural networks and human operators on the same task. By overcoming the fundamental limitations of computers, synaptic resistor circuits offer high-speed concurrent learning and inference, ultra-low power consumption, error correction, and agile adaptability for artificial intelligence systems.

Artificial intelligence (AI) draws its foundational concepts from the brain. Within the brain, intricate neurobiological circuits process vast amounts of incoming spike signals in parallel, generating synaptic currents that trigger output spikes in analog parallel mode[1–5]. Concurrently, the synaptic weights (conductance) can be dynamically modified by the processed signals through learning mechanisms like spike-time-dependent plasticity (STDP)[3,5]. The memories in the human brain can be continuously modified over time, enabling humans to navigate changes and respond effectively to novel situations[6,7]. In stark contrast, computers operating on the Turing model excel at executing pre-programmed inference algorithms, whether human-designed or derived from machine learning[8–18]. While AI systems, such as self-driving cars[14] and large language models[16], can surpass human performance in specific, well-defined domains, their inability to learn during inference renders them vulnerable to environmental changes, hardware errors, or task modifications. Although the human brain can also operate on fixed, pre-learned algorithms (Turing mode)[19,20], its unique ability to concurrently infer and learn, termed Super-Turing computing[19,20], distinguishes it from computers. For instance, computers can derive the algorithms to optimize wing shapes through off-site machine learning processes, but they cannot continuously adapt wing shapes like a bird in complex and rapidly changing aerodynamic environments while in flight[21–24]. Similarly, human intervention becomes necessary when self-driving cars encounter unforeseen scenarios[14,25]. Computer-based AI systems currently necessitate extensive data and energy-intensive off-site learning to broaden the operational domain, contrasting with the brain's efficient and continuous adaptation. Consequently, AI inference algorithms, trained on limited datasets, often struggle with the infinite complexity and unpredictable dynamics of the real world. Conversely, due to the "Turing constraint", computers require the expansion of learning domains for various conditions using "big data" and "deep-learning" technology, resulting in longer learning latency and higher energy consumption compared to the human brain[8–16]. As a result, the computationally demanding learning processes typically occur on large, energy-intensive remote computers to generate inference algorithms, which are subsequently deployed on power-constrained edge computers[8–13,16,25–28]. However, the AI inference algorithms derived from limited off-site training data often prove inadequate when applied to the unbounded complexity and unpredictable dynamics of real-world environments. Consequently, the computationally intensive learning processes often take place on large, power-hungry off-site computers to derive inference algorithms, which are then deployed on edge devices with power constraints[8–13,16,25–28]. The AI inference algorithms developed from finite training domains are limited in their effectiveness when applied to real-world environments with infinite complexity and unpredictable dynamic changes.

The super-Turing computing model has been postulated theoretically[19,20], but it was not established in terms of the concurrent

[1]Departments of Mechanical and Aerospace Engineering, Electrical and Computer Engineering, Materials Science and Engineering, California NanoSystems Institute, University of California, Los Angeles, CA, 90095, USA. [2]Department of Aerospace Engineering, University of Michigan, Ann Arbor, MI, 48104, USA. [3]Department of Aeronautics and Astronautics, Stanford University, Stanford, CA, 94305, USA. ✉e-mail: yongchen@seas.ucla.edu

inference and learning functionalities of neurobiological circuits. Neuromorphic computing circuits, built with digital transistors[10–13,25,26,28] or analog devices like floating-gate transistors[29,30], memristors[31–35], and phase-change memory resistors[36], aim to mimic biological neural networks for energy-efficient in-memory computing and in-situ learning. While learning algorithms like STDP have been successfully implemented in both digital and analog circuits, the resulting inference algorithms can only be executed after the learning phase is complete[10–13,25,26,28–36]. Consequently, these neuromorphic circuits are constrained to sequential learning and inference, preventing them from achieving super-Turing computing with simultaneous inference and learning[10–13,25,26,28–36]. The challenge remains: how can we design an electronic circuit capable of super-Turing computing—one that performs concurrent learning and inference, achieves high-energy efficiency, allows rapid learning, and adapts to dynamic environments?

In this article, we present a synaptic resistor (synstor)[37–41] circuit capable of operating in super-Turing mode, with concurrent inference and learning functionalities, to control a morphing wing in a wind tunnel —a complex and dynamic setting distinct from conventional AI test environments. We first introduce a super-Turing computing model based on a synstor circuit. We fabricated a synstor circuit, designed to mimic synapses by integrating inference, memory, and learning capabilities within each synstor to enable concurrent inference and learning in analog parallel mode. The synstor circuits accelerate learning speed, improve adaptability to dynamically changing environments, spontaneously correct device conductance errors, and reduce its operational conductance and power consumption by circumventing the need for sequential inference-learning and iterative learning-testing processes in the circuits of other neuromorphic devices. We then conducted experiments using the synstor circuit, humans, and a computer-based artificial neural network (ANN) to control a morphing wing. The objective was to minimize the drag-to-lift force ratio, reduce the fluctuation of the forces, and recover the wing from stall conditions by optimizing its shape in complex aerodynamic environments within a wind tunnel. Our results demonstrate that the synstor circuit and humans, operating in the super-Turing mode, outperform the ANN, operating in the Turing mode, in terms of learning speed, performance, power consumption, and adaptability to changing environments.

## Synstor circuit for intelligent systems

To emulate a neurobiological network, a circuit with $M$ input and $N$ output electrodes linked through $M \times N$ synstors is illustrated in Fig. 1a. When voltage pulses ($\mathbf{x}$) are applied to the input electrodes, they generate currents ($\mathbf{I}$) through the synstors at the output electrodes, implementing an inference algorithm:

$$\mathbf{I} = \mathbf{W}\,\mathbf{x} \tag{1}$$

where $\mathbf{W}$ represents the synstor conductance matrix. The excitatory or inhibitory currents ($\mathbf{I}$) stimulate or suppress voltage pulses ($\mathbf{y}$) from neuron and interface circuits, leading to changes in the system state ($\mathbf{s}$)—such as the configuration of a morphing wing (Fig. 1b). Sensors monitor the system state $\mathbf{s}$ (e.g., the lift-to-drag force ratio), and this feedback is converted back into input signals ($\mathbf{x}$) by the interface and neuron circuits. The control objective is to minimize the objective function $E = \frac{1}{2}s^2$.

In contrast to Turing-mode computing circuits, where inference algorithms (typically represented by $\mathbf{W}$) remain static during inference computation, the synaptic weights ($\mathbf{W}$) in our synstor circuit, similar to biological neural networks, can dynamically adapt through a concurrent correlative learning rule[4],

$$\dot{\mathbf{W}} = \alpha\, \mathbf{z} \otimes \mathbf{x} \tag{2}$$

where $\alpha$ denotes a learning coefficient, $\mathbf{z}$ denotes voltage pulses triggered at the output electrodes of the circuit, and $\mathbf{z} \otimes \mathbf{x}$ represents the outer product between $\mathbf{z}$ and $\mathbf{x}$ (i.e., $\frac{dw_{nm}}{dt} = \alpha\, z_n x_m$). The STDP learning rule[3,5] is also one of correlative learning rules[4] and can be represented by Eq. 2, where the temporal mean $\overline{z} = 0$ (Methods, Eq. 4), and the covariance between $z_n$ and $y_{n'}$, $<z_n, y_{n'}> = \eta_n\, \delta_{nn'}$ with $\eta_n \leq 0$ (Methods, Eq. 5). The concurrent execution of inference ($\mathbf{I} = \mathbf{W}\mathbf{x}$, Eq. 1) and learning ($\dot{\mathbf{W}} = \alpha\mathbf{z} \otimes \mathbf{x}$, Eq. 2) in a synstor circuit can result in the change rate of $\overline{E}$ (Methods):

$$\frac{d\overline{E}}{dt} \leq 0 \tag{3}$$

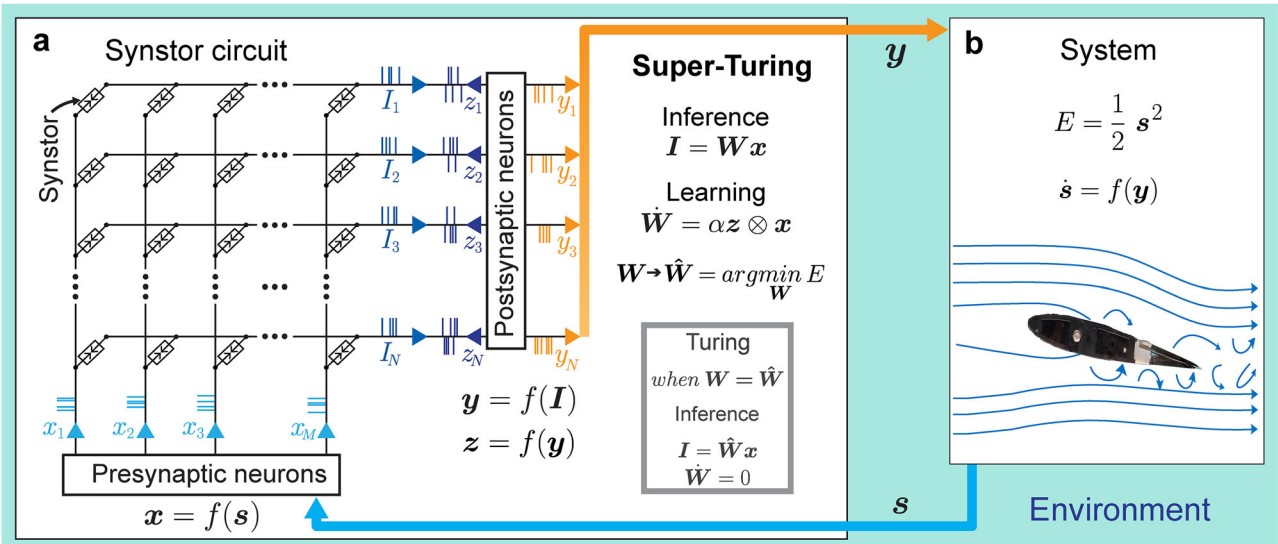

**Fig. 1 | A super-Turing synstor circuit for an intelligent morphing wing. a** A crossbar circuit is depicted, featuring synstors connecting presynaptic and postsynaptic neuron circuits. The input voltage pulses applied to the input electrode are represented by the vector $\mathbf{x}$. The output voltage pulses from the postsynaptic neuron form the vector $\mathbf{y}$, while the voltage pulses applied to the output electrode are given by the vector $\mathbf{z}$. The resulting current at the nth output electrode is represented by the vector i with elements in, which induces y and v from the postsynaptic neuron circuit. $\mathbf{I}$ induces y and z from the postsynaptic neuron circuit. **b** System states ($\mathbf{s}$) are detected by sensors and converted into input voltage pulses ($\mathbf{x}$) through the presynaptic neuron circuit. These input pulses drive the synstor circuit, which operates in super-Turing mode, concurrently executing an inference algorithm ($\mathbf{I} = \mathbf{W}\mathbf{x}$) and modifying its conductance matrix ($\mathbf{W}$) according to a learning rule ($\dot{\mathbf{W}} = \alpha\, \mathbf{z} \otimes \mathbf{x}$). The output ($\mathbf{y}$) from the synstor circuit controls actuators that modify the system states ($\mathbf{s}$). The goal is to minimize an objective function ($E = \frac{1}{2}\mathbf{s}^2$). When $E$ reaches its minimum, $\mathbf{W} = \hat{\mathbf{W}}$ and $\dot{\mathbf{W}} = 0$. Under this condition, the circuit only executes the inference algorithm $\mathbf{I} = \hat{\mathbf{W}}\mathbf{x}$ in Turing mode.

https://doi.org/10.1038/s44172-025-00437-y                                                    **Article**

Under this condition, $\overline{E}$ functions as a Lyapunov function for $\mathbf{W}$. When $\overline{E}$ reaches its minimum value, $\left(\frac{dE}{dt}\right) = 0$, and $\mathbf{W} = \hat{\mathbf{W}} = arg \min_W E$ remains unchanged ($\dot{\mathbf{W}} = 0$). The circuit operates in the Turing mode, executing the inference algorithm $\mathbf{I} = \hat{\mathbf{W}}\mathbf{x}$ (Fig. 1). When $\left(\frac{dE}{dt}\right) < 0$, $\dot{\mathbf{W}} \neq 0$, the circuit operates in the super-Turing mode, simultaneously performing inference and learning to adjust $\mathbf{W}$ toward $\hat{\mathbf{W}} = arg \min_W E$ and reduce $E$. A synstor circuit can operate as a conventional computing circuit in Turing mode, executing a fixed optimal inference algorithm. When the inference algorithm deviates from its optimal form due to environmental changes, task modifications, conductance errors, or other factors, the synstor circuit can spontaneously switch to super-Turing mode, simultaneously executing and optimizing the inference algorithm through learning to reduce $E$.

A synstor circuit (Fig. 2a) was fabricated following the methods detailed and illustrated in Fig. S1. Each synstor comprises a Si channel with Schottky contacts formed by Ti input and output electrodes via a metallic $TiSi_{0.9}$ layer[42]. Additionally, each synstor incorporates a vertical heterojunction stacked on the Si channel, consisting of a $SiO_2$ dielectric, a ferroelectric $Hf_{0.5}Zr_{0.5}O_2$ layer, and a $WO_{2.8}$ conductive reference electrode (Fig. 2b). During all electrical tests of this circuit (as detailed in the Methods), the reference electrodes were grounded. The electric tests of the synstor circuit (Methods) demonstrated its unique ability to execute both the inference ($I_n = \sum_m w_{nm}x_m$, Eq. 1) and learning ($\dot{w}_{nm} = \alpha z_n x_m$, Eq. 2) algorithms concurrently in super-Turing mode using the same type of signals. As shown in Fig. 2c, when voltage pulses ($x_m$) are applied to the $m^{th}$ input electrode, they induce currents that flow through the synstors to grounded output electrodes ($z_n = 0$), while the conductance of the synstors remains unchanged. The synstors connected to the grounded output electrodes simultaneously execute the inference ($I_n = \sum_m w_{nm}x_m$) and learning ($\dot{w}_{nm} = \alpha z_n x_m = 0$) algorithms. Concurrently, the conductance of other synstors experiencing positive or negative voltage pulses with the same amplitudes (i.e., $x_m = z_n > 0$ or $x_m = z_n < 0$) is modified according to the learning algorithm $\dot{w}_{nm} = \alpha z_n x_m$ with $\alpha < 0$ or $\alpha > 0$. As shown in experiments (Fig. S2a), device modeling (Methods), and an equivalent circuit for a synstor (Fig. 2b), when a single voltage pulse ($x_m$) is applied to the input electrode with respect to the grounded output electrode ($z_n = 0$), the voltage primarily drops across the Schottky contact between the input electrode and Si channel[42]. As a result, the $Hf_{0.5}Zr_{0.5}O_2$ layer beyond the Schottky contact experiences a small electric field that is insufficient to alter the ferroelectric domains or the synstor conductance. However, when voltage pulses ($x_m = z_n$) are simultaneously applied to the input and output electrodes of a synstor, there is no voltage drop across the Schottky contacts or the Si channel. Instead, the voltage primarily drops across the $Hf_{0.5}Zr_{0.5}O_2$ layer, generating a large electronic field that progressively switches the individual ferroelectric domains within the layer[43], thereby attracting or repelling the holes in the p-type Si channel, and increasing or decreasing the synstor conductance in analog mode. The oxygen vacancies with higher defect energy in the $Hf_{0.5}Zr_{0.5}O_2$ layer tended to diffuse toward the $WO_{2.8}$ reference electrode with lower defect energy, which effectively enhances the quality of the $Hf_{0.5}Zr_{0.5}O_2$ ferroelectric layer and improves device performance. Initially, the fabricated devices exhibited conductance variability, with an average conductance of 2.7 nS and a standard deviation of 2.1 nS. However, this variation was reduced to a standard deviation of 0.015 nS (Fig. S2) after tuning the devices to a target conductance value using a train of paired $x_m = z_n$ pulses with a duration of 10 $\mu s$ and an amplitude of $-4V$ (or $4V$). As a result, the synstor conductance can be precisely modified across 1000 analog conductance levels within a range of $1 - 60 nS$ (Fig. S2b) with a tuning accuracy of ~0.1 nS (Figs. S2c, d), $1.6 \times 10^{11}$ repetitive tuning cycles, and nonvolatile conductance retention time for over a year (Fig. S3e). During learning processes for various applications, we applied voltage pulses in the same manner to ensure precise tuning of device conductance and accurate execution of algorithms. The materials characterization and device properties of $Hf_{0.5}Zr_{0.5}O_2$ synstors will be detailed in a separate report. Compared with other neuromorphic circuits composed of analog devices such as floating-gate transistors[29,30], memristors[31–35], and phase-

change memory resistors[36], the synstor circuit has the unique capability to execute inference and learning algorithms concurrently in analog parallel mode using pulse signals ($\mathbf{x}$ and $\mathbf{z}$) with the same amplitudes (Fig. 2d). This enables the synstor circuit to operate in both super-Turing and Turing modes with very low conductance and power consumption, dynamically modifying the inference algorithm while executing it, adapting to dynamically changing environments, and correcting conductance errors in the circuit. However, as shown in the nonvolatile conductance retention test (Fig. S2e), after tuning the synstors to distinct analog conductance levels, their conductance was monitored over $10^6 s$ at room temperature. While the projected conductance levels remained distinct without overlap for a year, gradual shifts in conductance were observed over time. In Turing mode, where the synstor circuit operates without real-time learning, these conductance shifts can lead to computational errors. In contrast, in super-Turing mode, where real-time learning is enabled, the circuit can dynamically adjust the conductance to adapt to changing environments and correct errors, which makes synstor circuits more suited for super-Turing computing than traditional Turing computing.

## Morphing wing controlled by a synstor circuit

A morphing wing in a wind tunnel, as described in previous studies[44,45], was controlled by the synstor circuit (Fig. 3, Methods). The lift force ($F_L$) and drag force ($F_D$) on the wing were detected by strain gauges, with data processed by a computer (Fig. S3a). The sensing signals ($\mathbf{s}$), represented by the drag-to-lift force ratio ($s_1 = F_D/F_L$) and the magnitude of the fluctuation of the drag-to-lift force ratio ($s_2$), were transformed into voltage pulses ($x_1$ and $x_2$) and fed into the synstor circuit. The firing rate of $\mathbf{x}$ pulses increased monotonically with increasing amplitudes of $\mathbf{s}$. A $2 \times 2$ synstor circuit, composed of two input electrodes, two output electrodes, and four synstors, processed the input voltage pulses ($x_1$ and $x_2$) applied to the input electrodes. These pulses generated currents ($I_1$ and $I_2$) through the synstors at the output electrodes, implementing the inference algorithm ($\mathbf{I} = \mathbf{W}\mathbf{x}$, Eq. 1). The currents ($I_1$ and $I_2$) triggered voltage pulses ($y_1$ or $y_2$) through neuron circuits to increase or decrease the actuation voltage ($V_a$) applied on macro fiber composite (MFC) piezoelectric actuators[44,45], thereby adjusting the shape and states ($s_1$ and $s_2$) of the wing (Methods, Figs. S4, 5a, 6a). The $y_1$ or $y_2$ pulses increased or decreased the actuation voltage ($V_a$) applied on macro fiber composite (MFC) piezoelectric actuators[44,45], thereby altering the shape and states ($\mathbf{s}$) of the wing. The experiments were conducted in two different settings: one at a low angle of attack ($8°$) with the wing at pre-stall condition (Fig. S5a), and another at a high angle of attack ($18°$), exceeding the stall angle of the wing, thus introducing a chaotic aerodynamic environment at stall condition (Fig. S6a). The conductance matrix ($\mathbf{W}$) of the synstor circuit was initialized to random values before each experiment started to ensure that the circuit had no prior learning experience or predefined algorithm. When $\mathbf{y}$ pulses were triggered, $\mathbf{z}$ pulses satisfying the conditions $\overline{z}_n = 0$ (Eq. 4) and $\overline{z_n y_{n'}} = \eta_n \delta_{nn'}$ (Eq. 5) with $\eta_n \leq 0$ were also triggered at the output electrodes of the synstor circuit to modify $\mathbf{W}$ according to the correlative learning rule $\dot{\mathbf{W}} = \alpha \mathbf{z} \otimes \mathbf{x}$ (Eq. 2) in super-Turing mode. The objective of the experiments was to minimize the objective function $E = \frac{1}{2}\mathbf{s}^2$, representing both the drag-to-lift force ratio ($s_1 = F_D/F_L$) and its fluctuation ($s_2$), and to recover the wing from the stall state.

## Morphing wing controlled by human operators

In the experiments involving a morphing wing controlled by human operators (Fig. 3, Methods), operators with no prior knowledge of the wing or its control system visually received the sensing signals ($\mathbf{s}$) displayed on a computer monitor and were tasked with minimizing the objective function $E = \frac{1}{2}\mathbf{s}^2$ (Fig. S3b). These experiments were performed with the wing under the same pre-stall and stall conditions as those used for the synstor circuit. The $\mathbf{s}$ signals were processed by the human neurobiological circuits for inference ($\mathbf{I} = \mathbf{W}\mathbf{x}$, Eq. 1), prompting the operators to generate actuation signals ($\mathbf{y}$) by pressing two keys on a keyboard to adjust the actuation voltage $V_a$, thereby modifying the wing shape and its states $\mathbf{s}$. The firing rates of

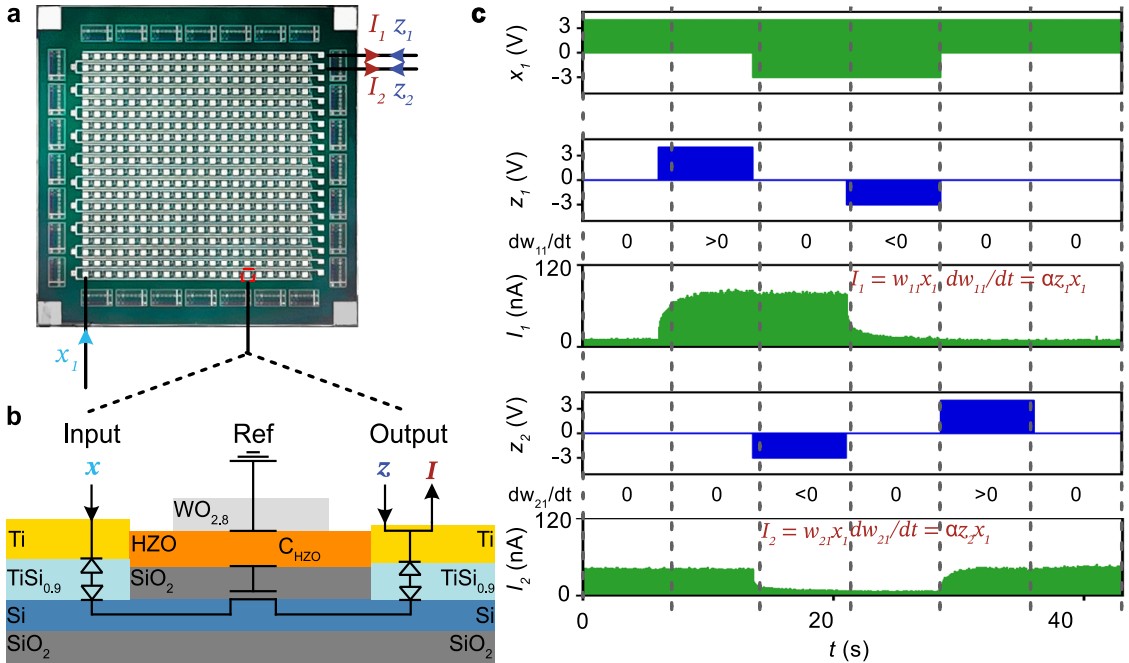

**Fig. 2 | A Hf$_{0.5}$Zr$_{0.5}$O$_2$-based synstor circuit. a** An optical image shows a 20 × 20 synstor crossbar, composed of 20 rows of Ti input electrodes and 20 columns of Ti output electrodes. **b** A schematic illustration of a synstor composed of a vertical heterojunction of a Si channel, a SiO$_2$ dielectric layer, a ferroelectric Hf$_{0.5}$Zr$_{0.5}$O$_2$ layer, and a WO$_{2.8}$ conductive reference (Ref) electrode. The Si channel connects with a TiSi$_{0.9}$ layer and Ti input and output electrodes. An equivalent circuit for the synstor is also shown, featuring diodes representing the Schottky contacts between the Si channel and Ti input/output electrodes via the TiSi$_{0.9}$ layer, a transistor formed by the Si channel and SiO$_2$ dielectric layer, and a capacitor ($C_{HZO}$) representing the Hf$_{0.5}$Zr$_{0.5}$O$_2$ ferroelectric layer beneath the WO$_{2.8}$ reference electrode. **c** Voltage pulses ($x_1$) applied on the first input electrode, voltage pulses applied on the first ($z_1$) and second ($z_2$) output electrodes, and currents flowing on the first ($I_1$) and second ($I_2$) output electrodes are shown against over time. The inference ($I_1 = w_{11}x_1$ and $I_2 = w_{21}x_1$) and learning algorithms ($dw_{11}/dt = \alpha z_1 x_1$ and $dw_{21}/dt = \alpha z_2 x_1$) are executed concurrently in the circuit in parallel analog mode under the various conditions, including (1) $x_1 = 4.2V$, $z_1 = z_2 = 0$, with $dw_{11}/dt = dw_{21}/dt = 0$; (2) $x_1 = 4.2V$, $z_1 = 4.2V$, $z_2 = 0$, with $dw_{11}/dt > 0$ and $dw_{21}/dt = 0$; (3) $x_1 = -2.1V$, $z_1 = 0$, $z_2 = -2.1V$, with $dw_{11}/dt = 0$, and $dw_{21}/dt < 0$; (4) $x_1 = -2.1V$, $z_1 = -2.1V$, $z_2 = 0$, with $dw_{11}/dt < 0$ and $dw_{21}/dt = 0$; and (5) $x_1 = 4.2V$, $z_1 = 0$, $z_2 = 4.2V$, with $dw_{11}/dt = 0$ and $dw_{21}/dt > 0$. **d** Comparative analysis of biological synapses[3], and analog neuromorphic devices including synstors (this work), floating-gate transistors[29,30], memristors[31–35], and phase-change memory resistors[36].

$y$ pulses corresponded to the duration of keystrokes (Figs. S5b, S6b). The conductance matrixes (**W**) of synapses in the human neurobiological circuits could be concurrently adjusted according to the STDP learning rule $\dot{\mathbf{W}} = \alpha \mathbf{z} \otimes \mathbf{x}$.

## Morphing wing controlled by ANN

In the experiments where a morphing wing was controlled by a state-of-the-art ANN with optimal structure and learning parameters (Fig. 3, Methods, Figs. S3c, S7), a computer received the sensing signals (**s**), executed the

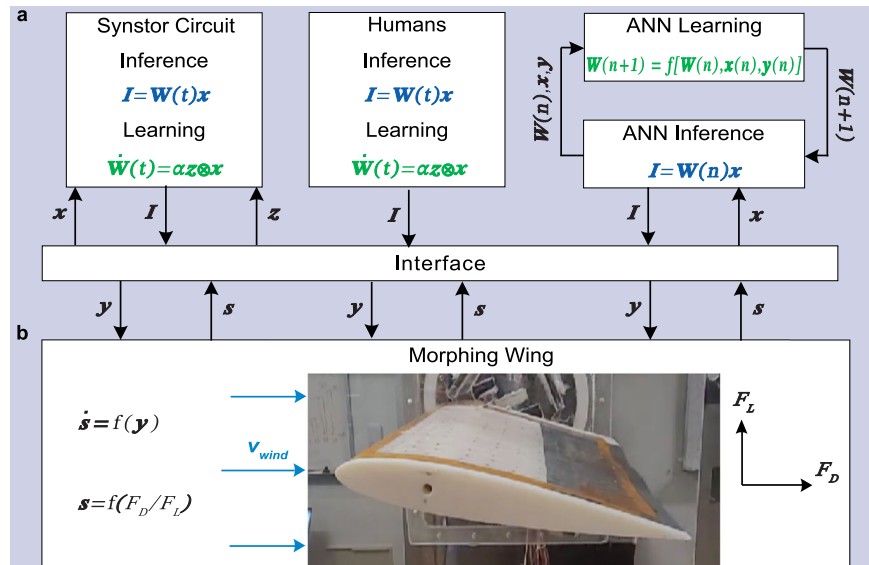

**Fig. 3 | System schematic showing a wing controlled by a synstor circuit, human operators, and a computer-based artificial neural network (ANN). a** A schematic shows the experimental settings where a synstor circuit (left), human operators (middle), and an ANN (right) all receive sensing signals of wing states (**s**), including the drag-to-lift force ratio ($s_1$) and its fluctuation magnitude ($s_2$), from the wing. The wing shape and its states (**s**) are modified by actuation signals (**y**). **b** An image displaying the morphing wing utilized in the wind tunnel experiments. During inference processes, the sensing signals **s** are converted to input signals (**x**), sequentially triggering output currents **I** according to the inference algorithm

$\mathbf{I} = \mathbf{W} \mathbf{x}$. In the synstor or human neurobiological circuits, the conductance matrixes (**W**) of synstors or synapses and inference algorithm $\mathbf{I} = \mathbf{W}(t)\mathbf{x}$ can be concurrently modified following the correlative learning rule $\dot{\mathbf{W}}(t) = \alpha\,\mathbf{z} \otimes \mathbf{x}$. In the sequential inference and learning processes of ANN, the inference data, including **x**, **y**, and $\mathbf{W}(n)$ from the $n^{th}$ inference episode are sent to a computer. Subsequently, $\mathbf{W}(n)$ is modified to $\mathbf{W}(n+1)$ in the $n^{th}$ learning episode according to a reinforcement learning algorithm $\mathbf{W}(n+1) = f[\mathbf{W}(n), \mathbf{x}(n), \mathbf{y}(n)]$. The inference algorithm $\mathbf{I} = \mathbf{W}(n+1)\mathbf{x}$ is then executed iteratively in the $(n+1)^{th}$ inference episode in the Turing mode.

inference algorithm ($\mathbf{I} = \mathbf{W} \mathbf{x}$, Eq. 1) within the ANN, and triggered actuation pulses (**y**) to adjust the actuation voltage $V_a$, thereby modifying the shape and states (**s**) of the wing. These experiments were performed with the wing under the same pre-stall and stall conditions as those used for the synstor circuit and human operators (Figs. S5c, S6c). To ensure a fair comparison, we used the policy gradient-based RL algorithm, Monte-Carlo Policy Gradient with baseline, as the benchmark. Since the actions were discrete, we did not use continuous action-based RL algorithms such as deep deterministic policy gradient (DDPG). Similar to the synstor trials, the synaptic weight matrices (**W**) in the ANN were initialized to random values before the learning experiment began. Due to the large data size, the time required to execute the learning algorithm was longer than that needed to execute the inference algorithm, therefore inference and learning were executed sequentially in Turing mode. In the offline learning process, the inference data, including **x**, **y**, and $\mathbf{W}(n)$ from the $n^{th}$ inference episode, were saved in the computer. The weight matrix $\mathbf{W}(n)$ was then modified to $\mathbf{W}(n+1)$ in the $n^{th}$ learning episode according to a reinforcement learning algorithm[46] (Methods, Supplementary Materials). The inference algorithm ($\mathbf{I} = \mathbf{W} \mathbf{x}$) was subsequently executed iteratively based on $\mathbf{W}(n+1)$ in the $(n+1)^{th}$ episode.

## Results

In the pre-stall condition with an 8° angle of attack, the synstor circuit, human operators, and ANN successfully learned to adjust the wing shape, minimizing the drag-to-lift force ratio ($s_1$) and the objective function $E = \frac{1}{2}\mathbf{s}^2$ (Fig. 4 and Supplementary Materials Movie S1). The fluctuation in the drag-to-lift force ratio ($s_2$) is inherently small under the pre-stall condition, and the change in $s_2$ remains negligible throughout the experimental process. When the wing was set in a stall condition with an 18° angle of attack, the synstor circuit and a few human operators successfully learned to adjust the wing shape, minimizing $s_1$, $s_2$, and $E$, thus recovering the wing state from stall (Fig. 5 and Supplementary Materials Movie S2). However, the ANN was unable to reduce $s_1$, $s_2$, or $E$ in sequential inference and learning trials under the stall condition.

$\frac{\partial E}{\partial y}$ can be extrapolated from the experimental data of $E$ and **y** using Eq. 10, Methods. $\frac{\partial E}{\partial y}$ and $\overline{E}$ are displayed versus time for synstor and human neurobiological circuits in both pre-stall and stall conditions (Figs. 4d, 5d). At the initial stage of the experiments, $\frac{\partial E}{\partial y} \neq 0$, thus the change rate in $\overline{E}$ due to learning, $\left(\frac{\partial \overline{E}}{\partial t}\right)_L = \overline{\frac{\partial E}{\partial W} \dot{W}} = \sum_n 2|\alpha|\eta_n \left(\frac{\partial E}{\partial E_x}\right)\left(\frac{\partial E_x}{\partial y_n}\right)^2 \left(\frac{\partial y_n}{\partial I_n}\right) = \sum_n 2|\alpha|\eta_n \left(\frac{\partial E}{\partial y_n}\right)\left(\frac{\partial E_x}{\partial y_n}\right)\left(\frac{\partial y_n}{\partial I_n}\right) < 0$ (Methods, Eq. 9), resulting in $\left(\frac{d\overline{E}}{dt}\right) < 0$ and $\dot{\mathbf{W}} \neq 0$. The synstor and human neurobiological circuits operate in super-Turing mode, simultaneously performing inference and learning to adjust **W** toward $\hat{\mathbf{W}} = \arg\min_W E$, reducing $\left|\frac{\partial E}{\partial y_n}\right|$ and $E$. At the late stage of the experiments, $\frac{\partial E}{\partial y} \approx 0$, $\left(\frac{\partial \overline{E}}{\partial t}\right)_L \approx 0$, $\overline{E}$ reaches its minimum value with $\left(\frac{d\overline{E}}{dt}\right) \approx 0$. Under this condition, $\dot{\mathbf{W}} \approx 0$ and $\mathbf{W} \approx \hat{\mathbf{W}}$, at which point the synstor and neurobiological circuits operate in Turing mode, executing only the inference algorithm $\mathbf{I} = \hat{\mathbf{W}}\mathbf{x}$. At the initial stage of the experiment for ANN in the pre-stall condition, $\frac{\partial E}{\partial y} \neq 0$ (Fig. 4d), $\frac{\partial E}{\partial W} \neq 0$ (Fig. S8a), and $\left(\frac{d\overline{E}}{dt}\right) < 0$. The computer sequentially executes the inference and learning algorithms in Turing mode, adjusting **W** toward $\hat{\mathbf{W}} = \arg\min_W E$ during learning, and reducing $\left|\frac{\partial E}{\partial y_n}\right|$ and $E$ during inference. At the late stage of the experiment, $\frac{\partial E}{\partial y}$, $\frac{\partial E}{\partial W}$, and $\left(\frac{d\overline{E}}{dt}\right)$ approach to zero, $\overline{E}$ reaches its minimum value, and $\mathbf{W} \approx \hat{\mathbf{W}}$. In the stall condition, the wing experiences a chaotic aerodynamic environment, resulting in larger fluctuations in $s_1$, $s_2$, or $E$ (Fig. 5) compared to the pre-stall condition (Fig. 4). Due to the chaotic changes in the environment, $\hat{\mathbf{W}} = \arg\min_W E$ varies dynamically, but **W** cannot be adjusted in real-time to adapt to these changes during inference. As a result, the sequential inference and learning fail to adjust **W** toward $\hat{\mathbf{W}}$ or reduce $E$ (Fig. S8b).

The $E - t$ curves can be best fitted by $E(t) = \left(E(0) - E_e\right) e^{-t/T_L} + E_e$ (Eq. 12) to extrapolate the initial learning time $T_L$ and the equilibrium objective function $E_e$ when $t \gg T_L$, and $\dot{E} \approx 0$. Under this equilibrium, both

**Fig. 4 | Experiment results and analysis for the wing in pre-stall condition with an 8° angle of attack. a** The drag-to-lift force ratios, $s_1 = F_D/F_L$, **b** the magnitude of the fluctuation of the drag-to-lift force ratio, $s_2$, and **c** the objective function, $E = \frac{1}{2}\mathbf{s}^2$ (arbitrary unit) and **d** $E$ and $\frac{\partial E}{\partial y_n}$ (arbitrary unit, Methods) for the wing controlled by a synstor circuit (left), a human (middle), and an ANN (right) are displayed versus time $t$ (gray and black lines), where $t$ represents concurrent inference and learning time in synstor and human neurological circuits, and the cumulative time for sequential inference and learning in the ANN. The average values of $s_1$ and $E$ are also shown versus time $t$ (red lines). The $E - t$ curves are best fitted by $E(t) = \left(E(0) - E_e\right) e^{-t/T_L} + E_e$ (green lines).

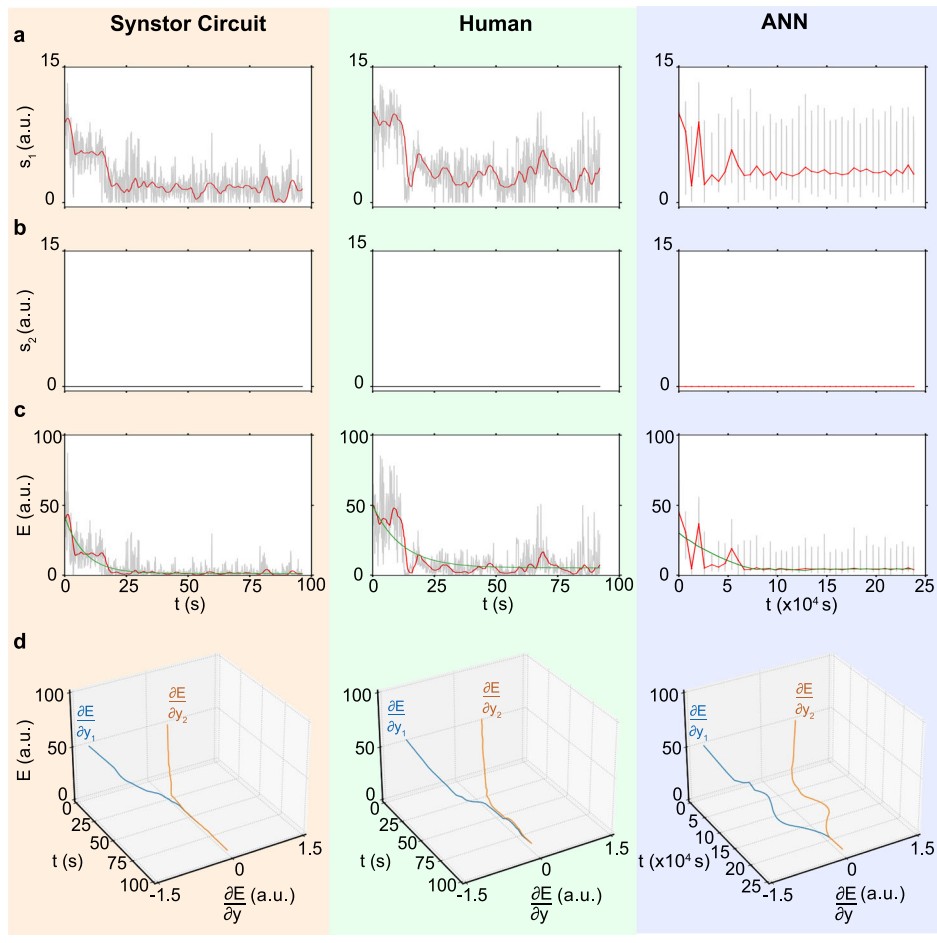

synstor circuits and human neurobiological circuits operate in Turing mode, with $\dot{\mathbf{W}} \approx 0$ and $\mathbf{W} \approx \hat{\mathbf{W}} = arg \min_W E$, and $\dot{E} \approx 0$. In the pre-stall condition with the wing at an 8° angle of attack, the average $T_L$ of the synstor circuit (4.6 s with a standard division $\sigma = 0.5$ s) in multiple trials is shorter than that of the human operators (16.8 s with $\sigma = 2.2$ s), and superior to that of the ANN (2656 s with $\sigma = 192$ s) (Fig. 6a). The average $E_e$ of the synstor circuit (1.4 a.u. with $\sigma = 0.2$) and humans (3.7 a.u. with $\sigma = 0.8$) in their multiple trials is superior to that of the ANN (4.3 a.u. with $\sigma = 0.9$) (Fig. 6a). Adaptability to the environment is represented by the successful rate in minimizing $E$ toward $E_e$ in multiple trials. In the pre-stall condition, the adaptability for the synstor circuit, human operators, and ANN is 100% with $\sigma = 0$ (Fig. 6a).

In the stall condition with the wing at an 18° angle of attack, the average $T_L$ of the synstor circuit (33.2s with $\sigma = 2.5$ s) across multiple trials is shorter than that of the human operators (55.8s with $\sigma = 7.5$ s), and superior to that of the ANN (> 34000 s) (Fig. 6b). The average $E_e$ of the synstor circuit (1.9 a.u. with $\sigma = 0.5$) across multiple trials is better than that of humans (30.0 a.u. with $\sigma = 7.1$) and the ANN (55.4 a.u. with $\sigma = 2.5$) (Fig. 6b). Adaptability to the environment is measured by the success rate in recovering the wing from stall and minimizing $E$ toward $E_e$ across multiple trials. The adaptability of the synstor circuit to the aerodynamic environment (100% with $\sigma = 0$) is better than that of humans (20% with $\sigma = 17\%$), and is superior to that of the ANN (0% with $\sigma = 0$) across their multiple trials (Fig. 6b). The power consumption of the synstor circuit (28 nW, Methods) for the concurrent execution of the inference and learning algorithms was eight orders of magnitude lower than the aggregate power consumption of the computer (5.0 W, Methods) executing the learning and inference algorithms sequentially. Estimating the power consumption of the human brain for inference and learning is difficult.

## Discussions and conclusions

We have created a synstor circuit model that mimics a neurobiological circuit by simultaneously executing inference ($\mathbf{I} = \mathbf{W}\,\mathbf{x}$, Eq. 1) and learning ($\dot{\mathbf{W}} = \alpha\, \mathbf{z} \otimes \mathbf{x}$, Eq. 2) algorithms in super-Turing mode. Theoretical analysis shows that this concurrent operation within a system can optimize the objective function of the system $E = \frac{1}{2}\mathbf{s}^2$ of the system with $\mathbf{s}$ representing the state of the system. When the inference algorithm deviates from its optimal form due to environmental shifts, conductance inaccuracies, or other influences, the synstor circuit actively corrects and optimizes it through simultaneous learning, driving the conductance matrix ($\mathbf{W}$) towards its ideal state ($\hat{\mathbf{W}} = arg \min_W E$) and minimizing the objective function $E$. Once the inference algorithm in the synstor circuit approaches its optimal configuration (i.e., $\mathbf{W} = \hat{\mathbf{W}}$), the circuit can operate in Turing mode, functioning as a conventional neuromorphic circuit executing a fixed, optimized inference algorithm ($\mathbf{I} = \hat{\mathbf{W}}\,\mathbf{x}$).

A synstor circuit was fabricated with a vertical stack comprising a Si channel, $SiO_2$ dielectric layer, ferroelectric $Hf_{0.5}Zr_{0.5}O_2$ layer, and a $WO_{2.8}$ reference electrode. Each synstor also features lateral Schottky contacts between the Si channel and $TiSi_{0.9}$/Ti input/output electrodes. Applying paired voltage pulses ($x_m$ and $z_n$) to these electrodes progressively tunes the ferroelectric domains in the $Hf_{0.5}Zr_{0.5}O_2$ layer, enabling analog conductance adjustment based on a correlative learning rule ($\dot{w}_{nm} = \alpha z_n x_m$). Conversely, a single input voltage pulse ($x_m$) induces current according to both an inference algorithm ($I_n = \sum_m w_{nm}x_m$) and the learning algorithm ($\dot{w}_{nm} = 0$).

Under this condition, the voltage primarily drops laterally across the Schottky junction and does not modify the ferroelectric domains within the $Hf_{0.5}Zr_{0.5}O_2$ layer, or the synstor conductance (i.e., $\dot{w}_{nm} = 0$ when $x_m \neq 0$ and $z_n = 0$) as per the learning rule ($\dot{w}_{nm} = \alpha z_n x_m$). Unlike other

**Fig. 5 | Experiment results and analysis for the wing in stall condition with an 18° angle of attack.** **a** The drag-to-lift force ratios, $s_1 = F_D/F_L$, **b** the magnitude of the fluctuation of the drag-to-lift force ratio, $s_2$, and **c** the objective function, $E = \frac{1}{2}\mathbf{s}^2$ (arbitrary unit), **d** $E$ and $\frac{\partial E}{\partial y_n}$ (arbitrary unit, Methods) for the wing controlled by a synstor circuit (left), a human (middle), and an ANN (right) are displayed versus time $t$ (gray and black lines), where $t$ represents concurrent inference and learning time in synstor and human neurological circuits, and the cumulative time for sequential inference and learning in the ANN. The average values of $s_1$ and $E$ are also shown versus time $t$ (red lines). The $E - t$ curves are best fitted by $E(t) = \left(E(0) - E_e\right)e^{-t/T_L} + E_e$ (green lines).

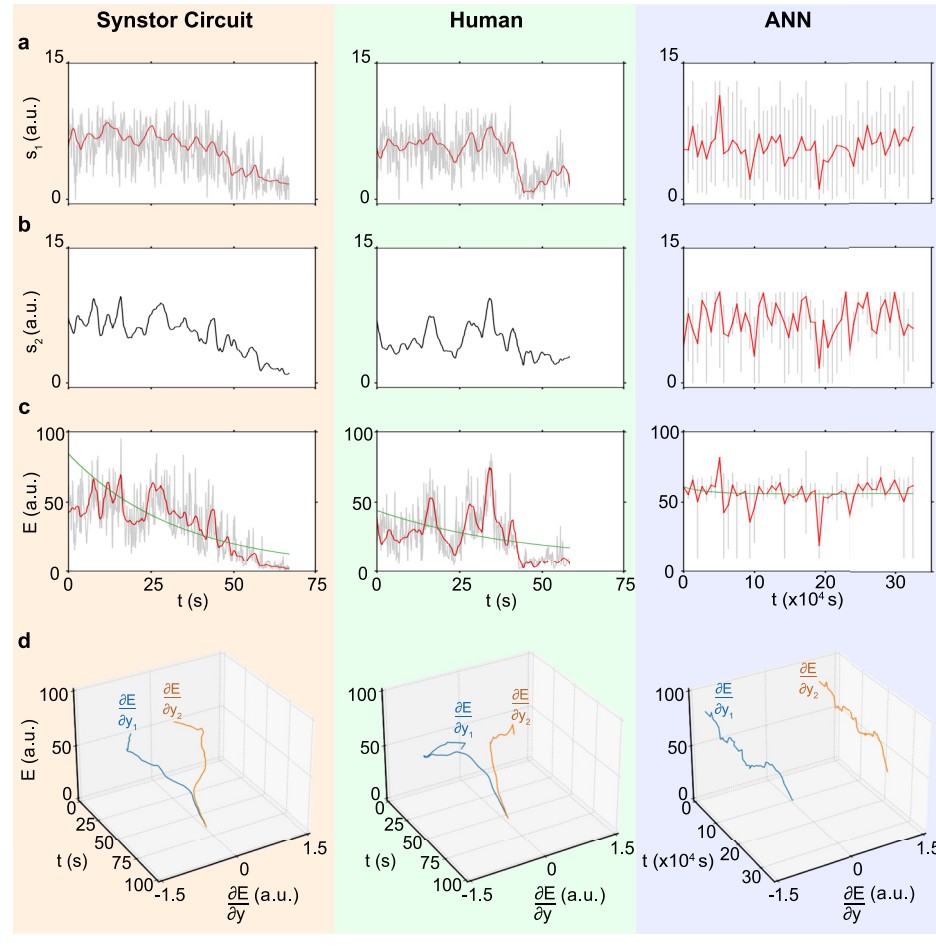

neuromorphic circuits, the synstor circuit has the unique capability to execute inference and learning algorithms concurrently in analog parallel mode, allowing the circuit to operate in both super-Turing and Turing modes. During inference algorithm execution, it can dynamically optimize the algorithm via learning, adapt to environmental changes, and correct the circuit conductance matrix. Unlike other neuromorphic circuits, this synstor circuit uniquely enables concurrent execution of inference and learning algorithms in analog parallel mode, allowing for dynamic algorithm optimization, adaptation to environmental changes, and correction of the conductance matrix of the circuit during inference, thus supporting both Super-Turing and Turing operation.

Experiments were conducted to control a morphing wing in a wind tunnel by a synstor circuit, humans, and a computer-based ANN in both pre-stall (8° angle of attack) and stall condition (18° angle of attack). The experimental objective was to minimize the drag-to-lift force ratio ($s_1 = F_D/F_L$), its fluctuation ($s_2$), and objective function $E = \frac{1}{2}\mathbf{s}^2$, recovering the wing from stall by optimizing the shape of a morphing wing. Without prior learning, a synstor circuit and humans executed learning and inference concurrently in Super-Turing mode, while the ANN executed inference and learning sequentially in Turing mode. In a synstor or neurobiological circuit, the conductance of each synstor or synapse can be dynamically adjusted and optimized in parallel analog mode to adapt to environmental changes. In contrast, an ANN cannot adjust its **W** matrix during inference in response to environmental changes; it requires sequential inference and learning to determine the statistically optimal **W** matrix across all conditions. Consequently, in the pre-stall condition, the synstor circuit and humans exhibited learning times ($T_L$) two orders of magnitude shorter than the ANN. In the stall condition, the synstor circuit and a few humans successfully optimized

the wing shape and adapted to the chaotic aerodynamic environment, recovering the wing from the stall. In contrast, the ANN failed to recover the wing from the stall. In stall condition, the wing faces a chaotic aerodynamic environment. During the inference process, both synstor and human neurobiological circuits can adjust and optimize their **W** matrices in response to these chaotic changes, allowing the wing to recover from the stall. In contrast, the ANN cannot adapt its **W** matrix during inference in response to environmental changes and fails to derive the statistically optimal **W** matrix across chaotic environments, leading to failure in recovering the wing from the stall. For the same reasons, the wing performance, measured by the post-learning equilibrium objective function ($E_e$), was superior for both the synstor circuit and humans compared to the ANN. The single-layer synstor circuit can execute learning and inference concurrently in real-time, dynamically optimizing its **W** matrix and inference algorithms, triggering optimal output actuation signals (**y**) to minimize the objective function ($E$). Conversely, the ANN and other neuromorphic circuits require additional time and energy for sequential data storage, learning algorithm execution, and data transfer between circuits. Moreover, the conductance ($< 60nS$) and power consumption ($< 100nW$) of synstors are lower than that of transistors ($< 1mS$ and $< 1mW$)[11–13,47], memristors ($\sim < 10mS$ and $< 1mW$)[31–35], and phase-change memory resistors ($< 10mS$ and $< 1mW$)[36]. Consequently, the power consumption of the synstor and neuron circuits (28 nW) for concurrent inference and learning is eight orders of magnitude lower than the aggregate power consumption (5.0 W) of the computer executing the learning and inference algorithms sequentially in the ANN. The speed to execute these algorithms in analog parallel mode scales linearly with the number of synstors ($MN$) in an $M \times N$ synstor circuit[37,38,41]. Synstor circuits offer a brain-inspired super-

**Fig. 6 | Comparative analysis of synstor circuit, human operators, and ANN.** The average learning time ($T_L$), equilibrium objective function ($E_e$), adaptability to the changing environment, and power consumption across multiple trials for the wing controlled by the synstor circuit (blue), human operators (green), and ANN (red) **a** in the pre-stall condition with the wing at an 8° angle of attack and **b** in the stall condition with the wing at an 18° angle of attack.

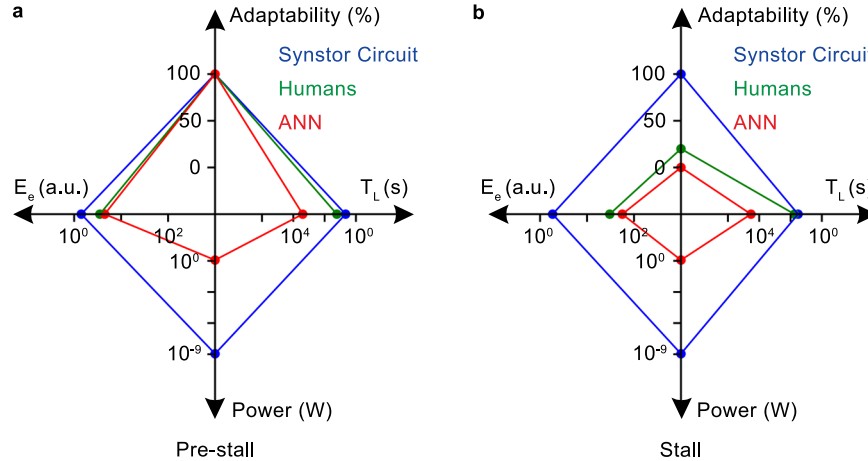

Turing computing platform for AI systems with extremely low power consumption, high-speed real-time learning and inference, self-correction of errors, and agile adaptability to dynamic complex environments.

## Methods
### Theoretical analysis of synstor circuits in super-Turing mode
The temporal mean of the $\mathbf{z}$ voltage pulses applied to the output electrode of the synstor circuit:

$$\overline{\mathbf{z}} = 0 \tag{4}$$

and the covariance between $z_n$ and $y_{n'}$,

$$\overline{z_n y_{n'}} = \,<z_n, y_{n'}> \, = \eta_n \delta_{nn'} \tag{5}$$

where $<z_n, y_{n'}> = \overline{(z_n - \overline{z_n})(y_{n'} - \overline{y_{n'}})} = \overline{z_n(y_{n'} - \overline{y_{n'}})} = \overline{z_n y_{n'}}$ due to $\overline{z_n} = 0$, $\delta_{nn'}$ denotes the Kronecker delta with $\delta_{nn'} = \begin{cases} 0 \text{ when } n' \neq n \\ 1 \text{ when } n' = n \end{cases}$, and $\eta_n$ represents a parameter with $\eta_n \leq 0$.

The learning rule observed in synapses within neurobiological circuits, known as STDP[3,5], can also be formulated as $\frac{d\mathbf{W}}{dt} = \alpha \, \mathbf{z} \otimes \mathbf{x}$ (Eq. 2) or $\frac{dw_{nm}}{dt} = \alpha z_n x_m$ with $z_n(t) = \begin{cases} A_- e^{(t-t_n^y)/\tau_-} \text{ when } t < t_n^y \\ -A_+ e^{-(t-t_n^y)/\tau_+} \text{ when } t \geq t_n^y \end{cases}$, where $t_n^y$ denotes the moment when a pulse ($y$) is triggered at the $n^{th}$ postsynaptic neuron, $A_+ > 0$ and $A_- > 0$ denote amplitude constants, and $\tau_+ > 0$ and $\tau_- > 0$ denote time constants. In STDP, $z$ also satisfies the conditions $\overline{z} = 0$ (Eq. 4) and $\overline{z_n y_{n'}} = \eta_n \delta_{nn'}$ (Eq. 5) with $\eta_n \geq 0$ for STDP and $\eta_n \leq 0$ for anti-STDP[5].

The change rate of objective function $E$ due to learning (modification of $\mathbf{W}$),

$$\left(\frac{\partial E}{\partial t}\right)_L = \frac{\partial E}{\partial \mathbf{W}} \dot{\mathbf{W}} = \sum_n \left(\frac{\partial E}{\partial E_x}\right)\left(\frac{\partial E_x}{\partial y_n}\right)\left(\frac{\partial y_n}{\partial I_n}\right) 2|\alpha| z_n E_x \tag{6}$$

where $E = \frac{1}{2}\sum_m s_m^2$, $E_x = \frac{1}{2}\sum_m x_m^2$, $\left(\frac{\partial E}{\partial t}\right)_L = \frac{\partial E}{\partial \mathbf{W}} \dot{\mathbf{W}} = \sum_{n,m} \left(\frac{\partial E}{\partial w_{nm}}\right) \dot{w}_{nm} = \sum_{n,m} \left(\frac{\partial E}{\partial E_x}\right)\left(\frac{\partial E_x}{\partial y_n}\right)\left(\frac{\partial y_n}{\partial I_n}\right)\left(\frac{\partial I_n}{\partial w_{nm}}\right) \dot{w}_{nm} = \sum_{n,m} \left(\frac{\partial E}{\partial E_x}\right)\left(\frac{\partial E_x}{\partial y_n}\right)\left(\frac{\partial y_n}{\partial I_n}\right) x_m (\alpha z_n x_m) = \sum_n \left(\frac{\partial E}{\partial E_x}\right)\left(\frac{\partial E_x}{\partial y_n}\right)\left(\frac{\partial y_n}{\partial I_n}\right) 2|\alpha| z_n E_x$, with $\frac{\partial I_n}{\partial w_{nm}} = x_m$ due to $I_n = \sum_m w_{nm} x_m$ (Eq. 1), and $\dot{w}_{nm} = \alpha z_n x_m$ (Eq. 2). $E_x$ and $y_n$ are discontinuous pulse functions and not differentiable; thus, $\frac{\partial E}{\partial E_x}, \frac{\partial E_x}{\partial y_n}$, and $\frac{\partial y_n}{\partial I_n}$ in Eq. 6 need to be derived through fitting experimental data. The change of $E_x$ within a

learning period can be expressed in a linear model as:

$$\delta E_x = \sum_{n'} \left(\frac{\partial E_x}{\partial y_{n'}}\right) \delta y_{n'} + \delta E_x^0 \tag{7}$$

where $\sum_{n'} \left(\frac{\partial E_x}{\partial y_{n'}}\right) \delta y_{n'}$ represents the change in $E_x$ due to $\delta y_{n'}$ with $\left(\frac{\partial E_x}{\partial y_{n'}}\right)$ as coefficients in the model, and $\delta E_x^0$ represents the part of $\delta E_x$ unrelated to $\delta y_{n'}$. By multiplying both sides of Eq. 7 by $z_n$ and then taking the temporal average over the learning period, we get

$$\overline{z_n E_x} = \left(\frac{\partial E_x}{\partial y_n}\right) \eta_n \tag{8}$$

where $\overline{z_n E_x} = \overline{z_n \delta E_x} = \sum_{n'} \left(\frac{\partial E_x}{\partial y_{n'}}\right)\overline{z_n \delta y_{n'}} + \overline{z_n}$ $\delta E_x^0 = \sum_{n'} \left(\frac{\partial E_x}{\partial y_{n'}}\right)$ $\overline{z_n y_{n'}} = \left(\frac{\partial E_x}{\partial y_{n'}}\right)\eta_n$, $\overline{z_n \delta E_x^0} = 0$, $\overline{z_n \delta E_x} = \overline{z_n(E_x - E_x(0))} = \overline{z_n E_x}$, $\overline{z_n \delta y_{n'}} = \overline{z_n(y_{n'} - y_{n'}(0))} = \overline{z_n y_{n'}}$ due to $\overline{z_n} = 0$ (Eq. 4), and $\sum_{n'} \left(\frac{\partial E_x}{\partial y_{n'}}\right)\overline{z_n y_{n'}} = \left(\frac{\partial E_x}{\partial y_{n'}}\right)\eta_n$ because $\overline{z_n y_{n'}} = \eta_n \delta_{nn'}$ (Eq. 5). The partial derivative, $\frac{\partial E}{\partial E_x}$, can be derived as a coefficient from a linear model, $\delta E = \left(\frac{\partial E}{\partial E_x}\right)\delta E_x + \delta E^0$, where $\left(\frac{\partial E}{\partial E_x}\right)\delta E_x$ represents the change in $E$ due to $\delta E_x$, and $\delta E^0$ represents the change of the part of $\delta E$ unrelated to $\delta E_x$. By multiplying both sides of the equation by $E_x - \overline{E_x}$ and then taking the temporal average over the learning period, $<E, E_x> \, = \, <\delta E, E_x> \, = \, \left(\frac{\partial E}{\partial E_x}\right)<\delta E_x, E_x> + <\delta E^0, E_x> = \left(\frac{\partial E}{\partial E_x}\right)<E_x, E_x>$, thus $\frac{\partial E}{\partial E_x} = \,<E, E_x> / <E_x, E_x>$. In the circuit, $E$ increases monotonically with increasing $E_x$, thus $\frac{\partial E}{\partial E_x} \geq 0$. The partial derivative, $\frac{\partial y_n}{\partial I_n}$, can be derived as a coefficient from a linear model, $\delta y_n = \left(\frac{\partial y_n}{\partial I_n}\right)\delta I_n + \delta y_n^0$, where $\left(\frac{\partial y_n}{\partial I_n}\right)\delta I_n$ represents the change in $y_n$ due to $\delta I_n$, and $\delta y_n^0$ represents the change of the part of $\delta y_n$ unrelated to $\delta I_n$. By multiplying both sides of the equation by $I_n - \overline{I_n}$ and then taking the temporal average over the learning period, $<y_n, I_n> \, = \, <\delta y_n, I_n> \, = \left(\frac{\partial y_n}{\partial I_n}\right)<\delta I_n, I_n> + <\delta y_n^0, I_n> = \left(\frac{\partial y_n}{\partial I_n}\right)<I_n, I_n>$, thus $\frac{\partial y_n}{\partial I_n} = \,<y_n, I_n> / <I_n, I_n>$. In the circuit, $y_n$ increases monotonically with increasing $I_n$, thus $\frac{\partial y_n}{\partial I_n} \geq 0$. Based on Eq. 6, the change rate of $\overline{E}$ due to learning within the learning period, $\left(\frac{\partial \overline{E}}{\partial t}\right)_L = \overline{\left(\frac{\partial E}{\partial t}\right)_L} = \sum_n \overline{\left(\frac{\partial E}{\partial E_x}\right)\left(\frac{\partial E_x}{\partial y_n}\right)} \left(\frac{\partial y_n}{\partial I_n}\right) 2|\alpha| z_n E_x = \sum_n \left(\frac{\partial E}{\partial E_x}\right)\left(\frac{\partial E_x}{\partial y_n}\right) \left(\frac{\partial y_n}{\partial I_n}\right) 2|\alpha| \overline{z_n E_x} = \sum_n 2|\alpha| \eta_n \left(\frac{\partial E}{\partial E_x}\right)\left(\frac{\partial E_x}{\partial y_n}\right)^2 \left(\frac{\partial y_n}{\partial I_n}\right)$, where

$\overline{z_n E_x} = \frac{\partial E_x}{\partial y_n} \eta_n$ (Eq. 8). During the learning process, $\eta_n \leq 0$ (Eq. 5), $\frac{\partial E}{\partial E_x} \geq 0$, and $\frac{\partial y_n}{\partial I_n} \geq 0$, thus

$$\left(\frac{\partial E}{\partial t}\right)_L = \overline{\frac{\partial E}{\partial \mathbf{W}} \dot{\mathbf{W}}} = \sum_n 2|\alpha|\eta_n \left(\frac{\partial E}{\partial E_x}\right)\left(\frac{\partial E_x}{\partial y_n}\right)^2 \left(\frac{\partial y_n}{\partial I_n}\right) \leq 0 \quad (9)$$

The overall change rate of $\overline{E}$, $\frac{d\overline{E}}{dt} = \left(\frac{\partial \overline{E}}{\partial t}\right)_L + \left(\frac{\partial \overline{E}}{\partial t}\right)_0$, where $\left(\frac{\partial \overline{E}}{\partial t}\right)_0$ represents the change rate of $\overline{E}$ unrelated to learning. When the system and learning parameters, such as $\alpha, \eta_n, \left|\frac{\partial E_x}{\partial y_n}\right|$, and $\frac{\partial y_n}{\partial I_n}$, are adjusted to meet the condition $\left(\frac{\partial \overline{E}}{\partial t}\right)_L \leq -\left(\frac{\partial \overline{E}}{\partial t}\right)_0$, then $\frac{d\overline{E}}{dt} \leq 0$ (Eq. 3).

## Synstor circuit fabrications and characterization

The synstor circuit was fabricated on a silicon-on-insulator wafer featuring a 220 nm p-doped Si layer on a 3-μm-thick buried silicon oxide layer. Ultraviolet (UV) photolithography and reactive ion etching (RIE; Oxford Plasmalab 80 Plus RIE) were used to fabricate 5 μm × 40 μm Si channels. After thermal oxidation to form a 3.5-nm-thick SiO2 layer and subsequent etching to define contact areas, 300-nm-thick Ti input/output electrodes were fabricated by UV photolithography, e-beam evaporation, and a lift-off process. The chip was annealed in forming gas (5% $H_2$ in $N_2$) at 460 °C for 30 min to form a titanium silicide layer sandwiched between the Si channel and Ti input/output electrodes. A 12.6-nm-thick $Hf_{0.5}Zr_{0.5}O_2$ layer was deposited on the chip by atomic layer deposition (Fiji Ultratech ALD) at 200 °C using tetrakis(dimethylamino)hafnium(IV) and tetrakis(dimethylamino)zirconium(IV) precursors. An 80-nm-thick W oxide layer was deposited on the $Hf_{0.5}Zr_{0.5}O_2$ layer by magnetron sputtering (Denton Discovery), and W oxide reference electrodes were patterned by lifting off a photoresist layer. A rapid thermal anneal (Modular Process Technology RTP-600xp) at 500 °C crystallized the $Hf_{0.5}Zr_{0.5}O_2$ layer and formed a $WO_{2.8}$ reference electrode. Finally, the $Hf_{0.5}Zr_{0.5}O_2$ was etched from the contact pads. The synstor features an active area of 200 μm². Based on our device simulations, an HfZrO-based synstor can be miniaturized to an active area of ~0.002 μm² using the 28 nm fabrication techniques for HfZrO-based ferroelectric transistors[48,49]. Structural and material composition profiles of the synstor chip were characterized using STEM (scanning transmission electron microscopy), EDX (energy-dispersive X-ray spectroscopy), and EELS (electron energy loss spectroscopy) analysis. EDX analysis was performed using JEOL JEM-2800 TEM operated at 200 kV. Atomic resolution STEM and EELS analysis was performed with a JEOL Grand ARM TEM operated at 300 kV with a spherical aberration corrector.

## Computer-aided device simulation

Based on the properties of synstors and their circuits, we have designed and simulated synstors by a technology computer-aided design (TCAD) simulator (Sentaurus Device, Synopsys). The simulator performed numerical calculations of the device physics by solving Poisson's equation describing the electrostatics and drift-diffusion carrier transport under a set of boundary conditions defined by the device structure. Quasi-stationary simulations were conducted under various voltage biases on the input/output electrodes of the synstors with respect to the grounded reference electrodes. The band diagrams of the synstors were extracted from these simulations, the electronic properties of the synstors were analyzed, and an equivalent circuit (Fig. 2b) was established to emulate these electronic properties.

## Electrical tests of synstor circuits

During the electric tests, the reference electrodes of the synstors were always grounded. Current-voltage characteristics were measured with a Keithley 4200 semiconductor parameter analyzer. The electrical voltage pulses applied to the input and output electrodes of the devices and circuits were generated by FPGA (National Instruments, cRIO-9063), computer-controlled modules (National Instruments, NI-9264), and a Tektronix AFG3152C waveform/function generator. Currents flowing through the synstors were measured by a semiconductor parameter analyzer, computer-controlled circuit modules (National Instruments, NI-9205 and NI-9403), and an oscilloscope (Tektronix TDS 3054B). Testing protocols were programmed (NI LabVIEW) and implemented in an embedded FPGA (Xilinx), a microcontroller, and a reconfigurable I/O interface (NI CompactRIO).

## Neuron circuits

Neuron circuits were designed in our laboratory to emulate the functions of biological neurons, and were fabricated in Taiwan Semiconductor Manufacturing Company (TSMC) according to our design. The structure and properties of the neuron circuit will be detailed in a separate report. Currents from excitatory (or inhibitory) synstors, I, are mirrored by transistors, leading the current $I$ to a capacitor of a transistor gate, thereby increasing (decreasing) its potential $V_m$. A leakage current, $I_b$, controlled by the voltage, $V_b$, reduces $V_m$. When $V_m$ reaches a threshold value, an output $y$ pulse is triggered. The neuron circuit fabricated by TSMC was tested by injecting the current $I$ into the neuron circuit, and measuring the voltage pulses output from the neuron circuit. As shown in Fig. S4, when $I < I_{th}$ (a threshold current), no pulse is output from the neuron circuit. When $I \geq I_{th}$, the frequency of the pulses output from the neuron circuit ($f_{out}$) increases with increasing $I$. The $I_{th}$ and $f_{out} - I$ relation can be adjusted by adjusting the control voltage ($V_b$). A digital circuit processes the $y$ pulses and generate $z$ pulses, with level shifters converting logic signals from the digital circuit to the desired voltage levels in the neuron circuit. As shown in Figs. S5a, S6a, when a $y_n$ pulse is triggered at $t = t_n$ from the $n^{th}$ neuron circuit, a 80-ms-wide $-2.1 V$ ($4.2 V$) $z_n$ pulses is triggered simultaneously at the $n^{th}$ ($n'^{th}$) output electrode connected with excitatory (inhibitory) synstors, and a 80-ms-wide $4.2 V$ ($-2.1 V$) $z_n$ pulse is triggered at the $n'^{th}$ ($n^{th}$) output electrode connected to the inhibitory (excitatory) synstors at $t = t_n + t_d$, with $t_d = 1.2s$. The $y$ and $z$ pulses satisfy the conditions $\overline{z} = 0$ (Eq. 4) and $\overline{z_n y_{n'}} = \eta_n \, \delta_{nn'}$ (Eq. 5) with $\eta_n \leq 0$. To minimize power consumption, all the analog transistors in the neuron circuit are operated in their sub-threshold region. The circuit simulations indicate that the average power consumption of the neuron circuit (including z pulse generation) is 0.27 nW.

## Morphing wing and wind tunnel

During reinforcement learning, the discrete action space comprised 500 distinct voltage levels used to either raise or lower the activation voltage applied to the piezoelectric actuators of the morphing wing. The wing is made of a NACA 0012 airfoil with 12-inch chord and 15-inch span rectangular profile. This morphing edge consisted of two macro fiber composite (MFC) piezoelectric actuators, each bonded to 0.001-inch stainless steel shims to create bending. Due to the antagonistic design of the morphing tail, two voltages opposite in sign but proportional in magnitude were supplied to the dual MFC system so that each MFC actuates the tail in the same direction. Therefore, although we only reported the actuation voltage, $V_a$, for one MFC, the second MFC received a separate voltage appropriately. Through a flexure box interface, the two MFC actuate antagonistically to smoothly and rapidly deflect the trailing edge and modify the camber of the airfoil, providing a multifunctional system acting as both skin and actuator[44,45]. An increase in actuation voltage causes the trailing edge of the morphing wing to deflect upward, while a decrease in voltage results in a downward deflection of the trailing edge. The design was scalable to multiple piezoelectric actuators along the spanwise edge (spanwise morphing trail edge) to achieve a continuous change in wing shape. The morphing wing experiments were performed in the open-loop wind tunnel facility at Stanford University with a square test section of 33 inch × 33 inch. The wind speed in the wind tunnel was set at 10.0 $m/s$. The lift ($F_L$) and drag

(F$_D$) forces on the wing were measured with strain gauges (OMEGA SGD-5/350-LY13) attached to a morphing wing mounting shaft.

## Experimental setup for a morphing wing controlled by a synstor circuit

As shown in Figs. S5a, 6a, the lift and drag forces on the wing were recorded by the strain gauges (OMEGA SGD-5/350-LY13), which was then connected to NI-9236 DAQ for analog to digital conversion and then processed by PC. The sensing signals, $s$, including the drag-to-lift force ratio ($s_1 = F_D/F_L$), and the magnitude of the fluctuation of the drag-to-lift force ratio ($s_2$) were converted to voltage pulses, $x$, with an amplitude of $-2.1\,V$ or $4.2\,V$ and a duration of $10\,ns$. The $x$ pulses were input to the synstor circuit via an interface circuit (FPGA, Xilinx, Kintex-7). The firing rate of $x$ pulses increased monotonically with increasing $s$. The $x$ signals generated currents via the synstor circuit by following the inference algorithm $\mathbf{I} = \mathbf{W}\,\mathbf{x}$ (Eq. 1), the currents, $\mathbf{I}$, flowed through the synstors to neuron/interface circuits, triggering the actuation pulses, $\mathbf{y}$, to modify the shape of the wing. When $y$ pulses were triggered, $z$ pulses satisfying the conditions $\bar{z} = 0$ (Eq. 4) and $\overline{z_n y_{n'}} = \eta_n \delta_{nn'}$ (Eq. 5) with $\eta_n \leq 0$ were also triggered at the output electrodes of the synstor circuit via the neuron circuits to modify $\mathbf{W}$ according to the learning rule $\dot{\mathbf{W}} = \alpha \mathbf{z} \otimes \mathbf{x}$ (Eq. 2) during the real-time learning process. The output pulses from the neuron circuits were converted to actuation voltage, $V_a$, via an interface circuit (FPGA, Xilinx, Kintex-7) to control the wing.

## Experimental setup for a morphing wing controlled by human operators

In the experiments with the morphing wing controlled by humans under the same environment of the synstor circuit experiments (Figs. S5b, S6b), human operators without any prior knowledge of the morphing wing and its control system visually received sensing signals ($s$) displayed on a computer monitor, and were instructed to minimize the $s$ or $E = \frac{1}{2}\mathbf{s}^2$ values. The human operators triggered actuation pulses $\mathbf{y}$ to change wing shape by pressing two keys in a keyboard. The firing rates of $y$ pulses were proportional to the keystroke times.

## Experimental setup for a morphing wing controlled by ANN

In the experiments with the morphing wing controlled by ANN under the same environment of the synstor circuit experiments (Figs. S5c, S6c), a Dell computer with Intel i7-8700 CPU received the $s$ signals, executed the inference algorithm ($\mathbf{I} = \mathbf{W}\,\mathbf{x}$, Eq. 1) in ANN with three layers of nine neurons and 20 synapses (Fig. S7), and triggered actuation pulses, $\mathbf{y}$, to change wing shape. We tested ANNs with various structures and learning parameters, selecting the optimal structures (nine neurons and 20 synapses) and learning parameters (a learning rate of $5 \times 10^{-9}\,a.u.$ and a discount factor of $0.99995\,a.u.$) with the shortest learning time and lowest objective function for the experiments. Before the experiment started, the synaptic weight matrix, $\mathbf{W}$, in the ANN was also set to random values, the same as for the synstor trials. Due to the large data size, the time for executing the learning algorithm was much longer than executing the inference algorithm. The ANN controller with synaptic weight matrixes $\mathbf{W}(n)$ controlled the wing for 30 s in the same environment as the synstor circuit in its $n^{th}$ round of trial, and the experimental data collected from this inference experiment was used to execute a policy gradient-based deep reinforcement learning algorithm[46] for about $600\,s$, modifying $\mathbf{W}(n)$ to $\mathbf{W}(n+1)$. The modified $\mathbf{W}(n+1)$ was sequentially sent back for the $(n+1)^{th}$ round of experiment to control the wing iteratively. In the case of pre-stall condition, after 40 iterative offline learning processes, the weights in the ANN gradually evolved from uncorrelated random values to stable correlated values, the ANN controller progressively learned to control the wing and the objective function $E = \frac{1}{2}\mathbf{s}^2$ was gradually decreased and minimized to a stable value. However, due to a highly chaotic environment during the stall condition, the ANN weights continuously oscillated and never stabilized in the experiments performed for 57 iterative offline learning processes.

Without real-time learning functionality, the computer was not able to dynamically optimize $\mathbf{W}$ in ANN to control the wing in the chaotic stall condition, and the wing failed to recover the wing from the stall and minimize the objective function $E = \frac{1}{2}\mathbf{s}^2$.

## Analysis of $\left(\frac{\partial E}{\partial y_n}\right)$ during learning processes

During the learning processes of the synstor circuits, humans, and ANN, $\frac{\partial E}{\partial y_n}$ can be derived from a linear model $\delta E = \sum_{n'}\left(\frac{\partial E}{\partial y_{n'}}\right)\delta y_{n'} + \delta E^0$, where $\sum_{n'}\left(\frac{\partial E}{\partial y_{n'}}\right)\delta y_{n'}$ represents the change in $E$ due to learning ($\delta y_{n'}$) with $\left(\frac{\partial E}{\partial y_{n'}}\right)$ as coefficients in the model, and $\delta E^0$ represents the part of $\delta E$ unrelated to $\delta y_{n'}$. By multiplying both sides of the equation by $y_n - \bar{y}_n$ and then taking the temporal average over a learning period, $<E, y_n> = <\delta E, y_n> = \sum_{n'}\left(\frac{\partial E}{\partial y_{n'}}\right) <\delta y_{n'}, y_n> + <\delta E^0, y_n> = \left(\frac{\partial E}{\partial y_n}\right)<y_n, y_n>$, where the covariance $<\delta E^0, y_n> = 0$, $<\delta y_{n'}, y_n> = <y_{n'}, y_n> = <y_n, y_n>\delta_{nn'}$, and $\sum_{n'}\left(\frac{\partial E}{\partial y_{n'}}\right)<\delta y_{n'}, y_n> = \sum_{n'}\left(\frac{\partial E}{\partial y_{n'}}\right)<y_n, y_n>\delta_{nn'} = \left(\frac{\partial E}{\partial y_n}\right)<y_n, y_n>$, thus

$$\left(\frac{\partial E}{\partial y_n}\right) = \ <E, y_n> / <y_n, y_n> \tag{10}$$

When $\frac{\partial E}{\partial y} \neq 0$, and $\frac{\partial E}{\partial \mathbf{W}} = \frac{\partial E}{\partial y}\frac{\partial y}{\partial \mathbf{W}} \neq 0$, the circuit operates in the super-Turing mode, simultaneously performing inference and learning to adjust $\mathbf{W}$ toward $\hat{\mathbf{W}} = \underset{W}{arg\ min}\ E$, while $\left|\frac{\partial E}{\partial \mathbf{W}}\right|$ gradually decreases toward 0. When $\overline{E}$ reaches its minimum value, $\frac{\partial E}{\partial y} = 0$, and $\frac{\partial E}{\partial \mathbf{W}} = \frac{\partial E}{\partial y}\frac{\partial y}{\partial \mathbf{W}} = 0$, $\dot{\mathbf{W}} = 0$, and $\mathbf{W} = \hat{\mathbf{W}} = \underset{W}{arg\ min}\ E$ remains unchanged, the circuit operates in the Turing mode (Figs. 4d, 5d).

## Analysis of $\left(\frac{\partial E}{\partial \mathbf{W}}\right)$ during ANN learning processes

During the ANN learning processes, $\frac{\partial E}{\partial \mathbf{W}}$ can be derived from a linear model $\delta E = \sum_{n', m'}\left(\frac{\partial E}{\partial w_{n'm'}}\right)\delta w_{n'm'} + \delta E^0$, where $\sum_{n', m'}\left(\frac{\partial E}{\partial w_{n'm'}}\right)\delta w_{n'm'}$ represents the change in $E$ due to learning ($\delta w_{n'm'}$), with $\left(\frac{\partial E}{\partial w_{n'm'}}\right)$ as coefficients in the model, and $\delta E^0$ represents the part of $\delta E$ unrelated to $\delta w_{nm}$. By multiplying both sides of the equation by $w_{nm} - \bar{w}_{nm}$ and then taking the temporal average over a learning period, $<E, w_{nm}> = <\delta E, w_{nm}> = \sum_{n', m'}\left(\frac{\partial E}{\partial w_{n'm'}}\right)<\delta w_{n'm'}, w_{nm}> + <\delta E^0, w_{nm}> = \left(\frac{\partial E}{\partial w_{nm}}\right)<w_{nm}, w_{nm}>$, where the covariance $<\delta E^0, w_{nm}> = 0$, $<\delta w_{n'm'}, w_{nm}> = <w_{n'm'}, w_{nm}> = <w_{nm}, w_{nm}>\delta_{nn'}\delta_{mm'}$, and $\sum_{n', m'}\left(\frac{\partial E}{\partial w_{n'm'}}\right)<\delta w_{n'm'}, w_{nm}> = \left(\frac{\partial E}{\partial w_{nm}}\right)<w_{nm}, w_{nm}>$, leading to:

$$\left(\frac{\partial E}{\partial w_{nm}}\right) = \ [<E, w_{nm}> / <w_{nm}, w_{nm}>] / <w_{nm}, w_{nm}> \tag{11}$$

During the ANN learning process of ANN, $\left(\frac{\partial E}{\partial w_{nm}}\right)$ can be derived from $E$ and $\mathbf{W}$ data based on Eq. 11 (Fig. S8). However, during the learning processes of the synstor and human neurobiological circuits, $\mathbf{W}$ was neither measured nor recorded, thus $\frac{\partial E}{\partial w_{nm}}$ cannot be derived.

## Analysis of the objective function $E$ during learning processes

During the learning processes of the synstor circuits, humans, and ANN, the average change rate of objective function, $\langle\dot{E}\rangle$, is a nonlinear function of time, but can be best fitted by a linear dynamic model $\langle\dot{E}\rangle = -(\langle E\rangle - E_e)/T_L$ and its solution over the learning processes,

$$\langle E(t)\rangle = \left(E(0) - E_e\right)e^{-t/T_L} + E_e \tag{12}$$

where the fitting parameter $T_L$ represents the average learning time, and $E_e$ represents the equilibrium objective function when $t \gg T_L$ and $\langle\dot{E}\rangle \approx 0$ (Figs. 4c, 5c). In this condition, both synstor circuits

and human neurobiological circuits operate in Turing mode, with $\dot{\mathbf{W}} \approx 0$ and $\mathbf{W} \approx \hat{\mathbf{W}} = \arg\min_W E$, after the learning processes conclude.

## Power consumption of inference and learning by the synstor circuit

During learning in the synstor circuit, conductance tuning was performed by applying a pair of voltage pulses with identical amplitudes to the input and output electrodes at the same time. These pulses charge the capacitor formed between the reference electrode and the silicon channel but did not drive current through the Si channel, unlike during inference. The average power consumption for learning in the synstor circuit can be estimated as: $P_L \approx c_T V_a^2 f_p$, where $c_T$ is the total capacitance of the synstors in the circuit, $V_a$ is the magnitude of pulses, and $f_p$ is the average frequency of the pulses applied for learning. In the learning process of the synstor circuit, the parameters are approximately, $c_T \approx 3.5 pF$, $V_a = 4.2V$, and $f_p \approx 0.6 Hz$, resulting in $P_L \approx 8.8 pW$.

During inference in the synstor circuit, voltage pulses were applied to the input electrodes of the synstors, while their output electrodes were held at ground potential. The average power consumption of the circuit for inference,

$$P = \mathbf{I} \otimes \mathbf{x} = (\mathbf{wx}) \otimes \mathbf{x} \approx w_T V_a^2 D_p \tag{13}$$

where $w_T$ denotes the total conductance of the synstors in the circuit, $V_a$ denotes the magnitude of pulses, $D_p$ denotes the average duty-cycle of the pulses. During the inference in the synstor circuit for the wing, $w_T \approx 40 nS$, $V_a = 4.2V$, and $D_p = 0.04$, thus $P \approx 28 nW$ (Fig. 6). The power consumption for learning is substantially lower than that required for inference. As a result, inference power primarily determines the overall power consumption of the synstor circuit.

## Power consumption of sequential inference and learning in ANN by the computer

According to analysis from Python toolkits "keras_flops" and "pyperf", the speeds for sequentially executing the inference and learning programs in ANN on the computer were estimated to be $1.24 KFLOPS$ and $2.0 GFLOPS$, respectively. With a computing energy efficiency of $0.4 GFLOPS/W$ [50], the power consumption for sequentially executing the inference and learning programs on the computer was $3.1 \mu W$ and $5.0 W$, respectively. The aggregate power consumption of the ANN for sequential inference and learning is $5.0 W$ (Fig. 6).

## Data availability
All data are available in the main text, Supplementary Information, or upon request from the corresponding author.

## Code availability
The source codes supporting this research are available on GitHub: https://github.com/Deo-Atharva/Morphing-Wing/tree/main] (https://github.com/Deo-Atharva/Morphing-Wing/tree/main.

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

## Acknowledgements

The authors acknowledge the support of this work by the Air Force Office of Scientific Research (AFOSR) under the programs, "Brain-Inspired Networks for Multifunctional Intelligent Systems in Aerial Vehicles (FA9550-19-0213)", "Center of Neuromorphic Computing under Extreme Environments (FA9550-24-1-0322)", and "Self-learning neuromorphic circuits of high-energy efficiency (FA9550-23-1-0638)".

## Author contributions

A.D. performed the wing experiments; A.D. and R.S. developed the code and system; A.D., J.L., D.G., Z.R., Y.H., and D.Q. designed, simulated, and fabricated the devices and circuits; K.PT.H. and D.J.I. designed the morphing wings, T.T. and F.C. designed the wing tunnel experiment, A.D. performed analysis; Y.C. initiated and guided the work.

## Competing interests

The authors declare no competing interests.
