## [Transparent Peer Review file · Communications Engineering]

Super-Turing Synaptic Resistor Circuits for Intelligent Morphing Wing

Corresponding Author: Professor Yong Chen

Version 0:

Reviewer comments:

Reviewer #1

(Remarks to the Author)

This manuscript introduces a synaptic resistor circuit that operates in Super-Turing mode. It emulates neurobiological systems by simultaneously learning and inference to outperform Turing mode in dynamically changing conditions. The HfZrO₂-based synstor circuit is fabricated and tested with morphing wing control tasks. The results show that the adaptability of the proposed circuits performs better than human and significantly outperforms ANN, and is also more energy efficient than ANN without sequential learning and inference. The idea is interesting, and the experiments are solid, but there are still some critical issues to be addressed.

1. What's the synaptic resistor circuit scope did the authors apply for the morphing wing control experiments? How many synstors?
2. What's the area of the proposed synstor? How is the scalability of the device?
3. The proposed synstor is reported to have very precise conductance. How is the process variation and its impact on algorithm accuracy? The extended data fig. 2 seems to be a result of fine-tuning, but is it feasible in real applications?
4. The authors adopted reinforcement learning as the ANN counterpart. Which RL algorithm did the authors applied? Some RL algorithms like DDPG are more suitable for continuous workloads. It is unclear whether the RL algorithm that the authors chose is the state-of-the-art for fair comparison.
5. The authors chose morphing wing control task as the experimental environment, which is not a very common testing case for ANN researchers. So similar as the last question, it's hard to tell if the results of proposed system are truly good or it's just the comparison with bad counterparts.
6. How's the scalability of the super-Turing mode algorithm? Could it be extended to multi-layer and perform stronger ability?

Reviewer #2

(Remarks to the Author)

The authors presented a paper titled " Super-Turing Synaptic Resistor Circuits for Intelligent Morphing Wing", in which the authors developed a synaptic resistor circuit that operates in Super-Turing mode, enabling concurrent learning and inference. Despite presenting a detailed results. My comment is:

The authors should indicate the trade-off of their approach

Version 1:

Reviewer comments:

Reviewer #1

(Remarks to the Author)

Most of my questions have been addressed. A few follow-up questions:

1. About tuning conductance to reduce variation, the conductance should keep changing during the super-tuning mode, so is the tuning applied all the time during training to control the morphing wing? Does its latency and energy get included in the overall consumption?

2. For the reinforcement learning method, what are the discrete actions in this scenario to control the morphing wing? like how many angles or directions?

Reviewer #2

(Remarks to the Author)
The paper can be accepted

Version 2:

Reviewer comments:

Reviewer #1

(Remarks to the Author)
I don't have more questions. The paper can be accepted.

HENRY SAMUELI SCHOOL OF ENGINEERING & APPLIED SCIENCE
MECHANICAL & AEROSPACE ENGINEERING DEPARTMENT
CALIFORNIA NANOSYSTEMS INSTITUTE
38-137G ENGINEERING IV BUILDING
LOS ANGELES, CALIFORNIA 90095-1597
FAX (310) 206-2302

Yong Chen, Professor

Tel: 310 206-2453 (Office/Fax)

E-mail: yongchen@seas.ucla.edu

February 16, 2025

Dear Editors and Reviewers,

We have revised the manuscript in response to the reviewers' suggestions, with all changes highlighted for your reference.

Reviewer #1

1. What's the synaptic resistor circuit scope did the authors apply for the morphing wing control experiments? How many synstors?

A 2×2 synstors circuit, composed of two input electrodes, two output electrodes, and four synstors, processed the input voltage pulses (x_1 and x_2) applied to the input electrodes. These pulses generated currents (I_1 and I_2) through the synstors at the output electrodes, implementing the inference algorithm ($I = W x$, Eq. 1). The currents (I_1 and I_2) triggered voltage pulses (y_1 or y_2) through neuron circuits to increase or decrease the actuation voltage (V_a) applied on macro fiber composite (MFC) piezoelectric actuators^{46,47}, thereby adjusting the shape and states (s_1 and s_2) of the wing (Methods, Extended Data Figs. 4, 5a, and 6a).

2. What's the area of the proposed synstors? How is the scalability of the device?

The device dimensions are detailed in the Methods section: “A Si channel measuring 5 μm in width and 40 μm in length was fabricated using reactive ion etching (RIE; Oxford Plasmalab 80 Plus RIE).” “An 80 nm-thick W oxide layer was deposited on the $\text{Hf}_{0.5}\text{Zr}_{0.5}\text{O}_2$ layer by magnetron sputtering (Denton Discovery),” “The synstor features an active area of 200 μm^2 . Based on our device simulations, an HfZrO-based synstor can be miniaturized to an active area of approximately 0.002 μm^2 using the 28 nm fabrication techniques for HfZrO-based ferroelectric transistors^{50,51}.”

3. The proposed synstor is reported to have very precise conductance. How is the process variation and its impact on algorithm accuracy? The extended data fig. 2 seems to be a result of fine-tuning, but is it feasible in real applications?

“The as-fabricated devices initially showed significant conductance variation, with an average conductance of 2.7 nS and a standard deviation of 2.1 nS. However, this variation was significantly reduced to a standard deviation of 0.015 nS (Extended Data Fig. 2) after tuning the devices to a target conductance value using a train of paired $x_m = z_n$ pulses with a duration of 10 μs and an amplitude of -4 V (or 4 V).” “During learning processes for various applications, we applied voltage pulses in the same manner to ensure precise tuning of device conductance and accurate execution of algorithms.”

4. The authors adopted reinforcement learning as the ANN counterpart. Which RL algorithm did the authors applied? Some RL algorithms like DDPG are more suitable for continuous workloads. It is unclear whether the RL algorithm that the authors chose is the state-of-the-art for fair comparison.

“In the experiments where a morphing wing was controlled by a state-of-the-art ANN with optimal structure and learning parameters (Fig. 3, Methods, Extended Data Figs. 3c and 7), a computer received the sensing signals (\mathbf{s}), executed the inference algorithm ($\mathbf{I} = \mathbf{W} \mathbf{x}$, Eq. 1) within the ANN, and triggered actuation pulses (\mathbf{y}) to adjust the actuation voltage V_a , thereby modifying the shape and states (\mathbf{s}) of the wing. These experiments were performed with the wing under the same pre-stall and stall conditions as those used for the synstor circuit and human operators (Extended Data Figs. 5c, and 6c). To ensure a fair comparison, we used the policy gradient-based RL algorithm, Monte-Carlo Policy Gradient with baseline, as the benchmark. Since the actions were discrete, we did not use continuous action-based RL algorithms such as Deep Deterministic Policy Gradient (DDPG). Similar to the synstor trials, the synaptic weight matrices (\mathbf{W}) in the ANN were initialized to random values before the learning experiment began. Due to the large data size, the time required to execute the learning algorithm was significantly longer than that needed to execute the inference algorithm, therefore inference and learning were executed sequentially in Turing mode. In the offline learning process, the inference data, including \mathbf{x} , \mathbf{y} , and $\mathbf{W}(n)$ from the n^{th}

inference episode, were saved in the computer. The weight matrix $W(n)$ was then modified to $W(n + 1)$ in the n^{th} learning episode according to a reinforcement learning algorithm⁴⁸ (Methods, Supplementary Materials). The inference algorithm ($I = W x$) was subsequently executed iteratively based on $W(n + 1)$ in the $(n + 1)^{th}$ episode.”

5. The authors chose morphing wing control task as the experimental environment, which is not a very common testing case for ANN researchers. So similar as the last question, it’s hard to tell if the results of proposed system are truly good or it’s just the comparison with bad counterparts.

“Computers following the Turing model⁸ can accurately execute arbitrary inference algorithms, but these algorithms must be pre-programmed by humans or derived from machine-learning processes⁹⁻¹⁹. With their accurate execution of the pre-defined inference algorithms, AI systems, such as self-driving cars¹⁵ and large language models¹⁷, can surpass human performance within their respective learning domains and common test environments. However, once set, the algorithms cannot be modified for learning during inference computations, This limitation is referred to herein as the “Turing constraint”. During inference computations, if the inference algorithms become invalid or incorrect due to environmental changes, hardware errors, task modifications, or other factors, it cannot be adjusted or corrected, rendering it susceptible to failure under these conditions. The human brain can perform computations in Turing mode based on pre-learned algorithms that remain fixed during the inference computations^{20,21}, but the ability to simultaneously infer and learn, referred to as the Super-Turing computing mode^{20,21}, sets the brain apart from computers. A neurobiological circuit can process signals based on its synaptic weights while simultaneously adjusting these weights^{3,5}, enabling the modification, optimization, and correction of its inference algorithms in response to dynamic changes during the inference process²². For instance, computers can derive the algorithms to optimize wing shapes through off-site machine learning processes, but they cannot continuously adapt wing shapes like a bird in complex and rapidly changing aerodynamic environments while in flight²³⁻²⁶. When self-driving cars encounter unexpected environments beyond their pre-learning domains, human drivers must intervene, taking control and devising solutions using their concurrent inference and learning abilities in the new environments^{15,27}. Conversely, due to the “Turing constraint”, computers require the expansion of learning domains for various conditions using ‘big data’ and ‘deep-learning’ technology, resulting in significantly longer learning latency and higher energy consumption compared to the human brain⁹⁻¹⁷. Consequently, the computationally intensive learning processes often take place on large, power-hungry off-site computers to derive inference algorithms, which are then deployed on edge devices with power constraints^{9-14,17,27-30}. The AI inference algorithms developed from finite training domains are limited in their effectiveness when applied to real-world environments with infinite complexity and unpredictable dynamic changes.”

“In this article, we present a synaptic resistor (synstor)³⁹⁻⁴³ circuit capable of operating in Super-Turing mode, with concurrent inference and learning functionalities, to control a morphing wing in a wind tunnel — a complex and dynamic setting distinct from conventional AI test environments.” “Experiments were conducted to control a morphing wing in a wind tunnel by a synstor circuit, humans, and a computer-based ANN in both pre-stall (8° angle of attack) and stall condition (18° angle of attack). The experimental objective was to minimize the drag-to-lift force ratio ($s_1 = F_D/F_L$), its fluctuation (s_2), and objective function $E = \frac{1}{2} \mathbf{s}^2$, recovering the wing from stall by optimizing the shape of a morphing wing. Without prior learning, a synstor circuit and humans executed learning and inference concurrently in Super-Turing mode, while the ANN executed inference and learning sequentially in Turing mode. In a synstor or neurobiological circuit, the conductance of each synstor or synapse can be dynamically adjusted and optimized in parallel analog mode to adapt to environmental changes. In contrast, an ANN cannot adjust its \mathbf{W} matrix during inference in response to environmental changes; it requires sequential inference and learning to determine the statistically optimal \mathbf{W} matrix across all conditions. Consequently, in the pre-stall condition, the synstor circuit and humans exhibited learning times (T_L) two orders of magnitude shorter than the ANN. In the stall condition, the synstor circuit and a few humans successfully optimized the wing shape and adapted to the chaotic aerodynamic environment, recovering the wing from the stall. In contrast, the ANN failed to recover the wing from stall. In stall condition, the wing faces a chaotic aerodynamic environment. During the inference process, both synstor and human neurobiological circuits can adjust and optimize their \mathbf{W} matrices in response to these chaotic changes, allowing the wing to recover from the stall. In contrast, the ANN cannot adapt its \mathbf{W} matrix during inference in response to environmental changes and fails to derive the statistically optimal \mathbf{W} matrix across chaotic environments, leading to failure in recovering the wing from the stall. For the same reasons, the wing performance, measured by the post-learning equilibrium objective function (E_e), was significantly superior for both the synstor circuit and humans compared to the ANN. The single-layer synstor circuit can execute learning and inference concurrently in real-time, dynamically optimizing its \mathbf{W} matrix and inference algorithms, triggering optimal output actuation signals (\mathbf{y}) to minimize the objective function (E). Conversely, the ANN and other neuromorphic circuits require additional time and energy for sequential data storage, learning algorithm execution, and data transfer between circuits. Moreover, the conductance ($< 60 \text{ nS}$) and power consumption ($< 100 \text{ nW}$) of synstors are significantly lower than that of transistors ($< 1 \text{ mS}$ and $< 1 \text{ mW}$)^{12-14,49}, memristors ($\sim < 10 \text{ mS}$ and $< 1 \text{ mW}$)³³⁻³⁷, and phase change memory resistors ($< 10 \text{ mS}$ and $< 1 \text{ mW}$)³⁸. Consequently, the power consumption of the synstor and neuron circuits (28 nW) for concurrent inference and learning is eight orders of magnitude lower than the aggregate power consumption (5.0 W) of the computer executing the learning and inference algorithms sequentially in the ANN.” “Synstor circuits offer a brain-inspired Super-Turing computing platform for AI systems with extremely

low power consumption, high-speed real-time learning and inference, self-correction of errors, and agile adaptability to dynamic complex environments.”

Experiments were conducted to control a morphing wing in a wind tunnel by a synstor circuit, humans, and a computer-based ANN in both pre-stall (8° angle of attack) and stall condition (18° angle of attack). The experimental objective was to minimize the drag-to-lift force ratio ($s_1 = F_D/F_L$), its fluctuation (s_2), and objective function $E = \frac{1}{2} \mathbf{s}^2$, recovering the wing from stall by optimizing the shape of a morphing wing. Without prior learning, a synstor circuit and humans executed learning and inference concurrently in Super-Turing mode, while the ANN executed inference and learning sequentially in Turing mode. In a synstor or neurobiological circuit, the conductance of each synstor or synapse can be dynamically adjusted and optimized in parallel analog mode to adapt to environmental changes. In contrast, an ANN cannot adjust its \mathbf{W} matrix during inference in response to environmental changes; it requires sequential inference and learning to determine the statistically optimal \mathbf{W} matrix across all conditions. Consequently, in the pre-stall condition, the synstor circuit and humans exhibited learning times (T_L) two orders of magnitude shorter than the ANN. In the stall condition, the synstor circuit and a few humans successfully optimized the wing shape and adapted to the chaotic aerodynamic environment, recovering the wing from the stall. In contrast, the ANN failed to recover the wing from stall. In stall condition, the wing faces a chaotic aerodynamic environment. During the inference process, both synstor and human neurobiological circuits can adjust and optimize their \mathbf{W} matrices in response to these chaotic changes, allowing the wing to recover from the stall. In contrast, the ANN cannot adapt its \mathbf{W} matrix during inference in response to environmental changes and fails to derive the statistically optimal \mathbf{W} matrix across chaotic environments, leading to failure in recovering the wing from the stall. For the same reasons, the wing performance, measured by the post-learning equilibrium objective function (E_e), was significantly superior for both the synstor circuit and humans compared to the ANN. The single-layer synstor circuit can execute learning and inference concurrently in real-time, dynamically optimizing its \mathbf{W} matrix and inference algorithms, triggering optimal output actuation signals (\mathbf{y}) to minimize the objective function (E). Conversely, the ANN and other neuromorphic circuits require additional time and energy for sequential data storage, learning algorithm execution, and data transfer between circuits. Moreover, the conductance and power consumption of synstors ($< 60 \text{ nS}$ and $< 100 \text{ nW}$) are significantly lower than that of transistors ($< 1 \text{ mS}$ and $< 1 \text{ mW}$)^{12-14,49}, memristors ($\sim < 10 \text{ mS}$ and $< 1 \text{ mW}$)³³⁻³⁷, and phase change memory resistors ($< 10 \text{ mS}$ and $< 1 \text{ mW}$)³⁸. Consequently, the power consumption of the synstor and neuron circuits (28 nW) for concurrent inference and learning is eight orders of magnitude lower than the aggregate power consumption (5.0 W) of the computer executing the learning and inference algorithms sequentially in the ANN. To execute the generalized learning rule ($\dot{\mathbf{W}} = \alpha \mathbf{z} \otimes \mathbf{x}$) by applying various \mathbf{z} pulses, different learning

algorithms in all three major machine learning paradigms (unsupervised, supervised, and reinforcement learning)⁴ can be implemented in multi-layer synstor circuits. Synstor circuits offer a brain-inspired Super-Turing computing platform for AI systems with extremely low power consumption, high-speed real-time learning and inference, self-correction of errors, and agile adaptability to dynamic complex environments.

6. How's the scalability of the super-Turing mode algorithm? Could it be extended to multi-layer and perform stronger ability?

“We developed a synstor circuit model that emulates a neurobiological circuit, simultaneously performing inference ($\mathbf{I} = \mathbf{W} \mathbf{x}$, Eq. 1) and learning ($\dot{\mathbf{W}} = \alpha \mathbf{z} \otimes \mathbf{x}$, Eq. 2) algorithms in Super-Turing mode.” “The speed to execute these algorithms in analog parallel mode scales linearly with the number of synstors (MN) in an $M \times N$ synstor circuit^{39,40,43}. By applying different \mathbf{z} pulses to implement the generalized learning rule ($\dot{\mathbf{W}} = \alpha \mathbf{z} \otimes \mathbf{x}$), multi-layer synstor circuits can support various learning algorithms across all three major machine learning paradigms: unsupervised, supervised, and reinforcement learning⁴.”

Reviewer #2

The authors should indicate the trade-off of their approach

“However, as shown in the nonvolatile conductance retention test (Extended Data Fig. 2e), after tuning the synstors to distinct analog conductance levels, their conductance was monitored over 10^6 s at room temperature. While the projected conductance levels remained distinct without overlap for a year, gradual shifts in conductance were observed over time. In Turing mode, where the synstor circuit operates without real-time learning, these conductance shifts can lead to computational errors. In contrast, in Super-Turing mode, where real-time learning is enabled, the circuit can dynamically adjust the conductance to adapt to changing environments and correct errors, which makes synstor circuits more suited for Super-Turing computing than traditional Turing computing.”

Your review of this article will be highly appreciated.

Sincerely,

Yong Chen

1. About tuning conductance to reduce variation, the conductance should keep changing during the super-turing mode, so is the tuning applied all the time during training to control the morphing wing? Does its latency and energy get included in the overall consumption?

During the Super-Turing mode, the conductance of synstors continuously evolves. “During learning in the synstor circuit, conductance tuning was performed by applying a pair of voltage pulses with identical amplitudes to the input and output electrodes at the same time. Conductance tuning is performed by applying a pair of voltage pulses with identical amplitudes to the input and output electrodes at the same time. These pulses charge the capacitor formed between the reference electrode and the silicon channel but do not drive current through the Si channel, unlike during inference. The average power consumption for learning in the synstor circuit can be estimated as: $P_L \approx c_T V_a^2 f_p$, where c_T is the total capacitance of the synstors in the circuit, V_a is the magnitude of pulses, and f_p is the average frequency of the pulses applied for learning. In the learning process of the synstor circuit, the parameters are approximately, $c_T \approx 3.5 \text{ pF}$, $V_a = 4.2 \text{ V}$, and $f_p \approx 0.6 \text{ Hz}$, resulting in $P_L \approx 8.8 \text{ pW}$.” “The power consumption for learning is substantially lower than that required for inference. As a result, inference power primarily determines the overall power consumption of the synstor circuit.”

2. For the reinforcement learning method, what are the discrete actions in this scenario to control the morphing wing? like how many angles or directions?

“During reinforcement learning, the discrete action space comprised 500 distinct voltage levels used to either raise or lower the activation voltage applied to the piezoelectric actuators of the morphing wing. The edge of the morphing wing consisted of two macro fiber composite (MFC) piezoelectric actuators, each bonded to 0.001-inch stainless steel shims to create bending. Due to the antagonistic design of the morphing tail, two voltages opposite in sign but proportional in magnitude were supplied to the dual MFC system so that each MFC actuates the tail in the same direction. Therefore, although we only reported the actuation voltage, V_a , for one MFC, the second MFC received a separate voltage appropriately. Through a flexure box interface, the two MFC actuate antagonistically to smoothly and rapidly deflect the trailing edge and modify the camber of the airfoil, providing a multifunctional system acting as both skin and actuator^{45,46}. An increase in actuation voltage causes the trailing edge of the morphing wing to deflect upward, while a decrease in voltage results in a downward deflection of the trailing edge.”